# CALIBRATED PHYSICS-INFORMED UNCERTAINTY QUANTIFICATION

## ABSTRACT

Neural PDEs have emerged as inexpensive surrogate models for numerical PDE solvers. While they offer efficient approximations, they often lack robust uncertainty quantification (UQ), limiting their practical utility. Existing UQ methods for these models typically have high computational demands and lack guarantees. We introduce a novel framework for calibrated physics-informed uncertainty quantification to address these limitations. Our approach leverages physics residual errors as a nonconformity score within a conformal prediction (CP) framework. This enables data-free, model-agnostic, and statistically guaranteed uncertainty estimates. Our framework utilises convolutional layers as finite difference stencils for gradient estimation, our framework provides inexpensive coverage bounds for the violation of conservation laws within model predictions. In our experiments, we utilise CP to obtain marginal coverage for each cell and joint coverage over the entire prediction domain of various PDEs.

## 1 INTRODUCTION

Numerical Partial Differential Equation (PDE) solvers are crucial for scientific simulations across various fields (Danabasoglu et al., 2020; Giudicelli et al., 2024), but they often incur significant computational costs and carbon footprints (Horwitz, 2024). Machine learning offers an efficient alternative for approximating numerical simulations (Bertone et al., 2019; Karniadakis et al., 2021), finding applications in weather modelling (Lam et al., 2023; Kurth et al., 2023), fluid dynamics (Jiang et al., 2020; Pfaff et al., 2021), and nuclear fusion (Poels et al., 2023; Carey et al., 2024; Gopakumar & Samaddar, 2020). Unlike traditional numerical solvers, which may suffer from numerical instabilities, neural PDE solvers always produce an output. However, their accuracy and usefulness are not assured. They may violate system physics and confidently assert incorrect solutions due to various errors (Gopakumar et al., 2023a). Existing uncertainty quantification methods for neural PDE solvers often lack coverage guarantees (Zou et al., 2024), require expensive simulation data (Gopakumar et al., 2024a), or necessitate model modifications (Abdar et al., 2021).

To address these challenges, we propose a framework that leverages PDE residuals over neural PDEs to provide uncertainty estimates with guarantees. Our approach, outlined in fig. 1, takes obtaining predictions from the neural PDE solver, evaluates the associated Physics Residual Errors (PRE), and performs calibration using Conformal Prediction (CP) with both marginal (univariate) and joint (multivariate) formulations. This method provides statistically valid, guaranteed coverage within the residual space (Vovk et al., 2005), offering error bounds based on the violation of physical conservation laws. The framework is model-agnostic, requires no additional data, and is physics-informed. Through this work, we explore the (over)confidence issue of neural PDE solvers(Zou et al., 2024). This novel approach to uncertainty quantification can potentially be extended to ODEs and other systems where model performance can be expressed as a residual. Our contributions are:

- **Physics Residual Error as a nonconformity score for CP:** We introduce a novel nonconformity metric for CP over the residual space. Our approach is data-free and yields input-independent prediction sets while relaxing exchangeability restrictions.
- **Marginal and Joint CP:** Our approach provides guaranteed coverage bounds both marginally (per dimension) and jointly (entire prediction domain), enabling the identification of regions of volatile predictions and rejection sampling across predictions.

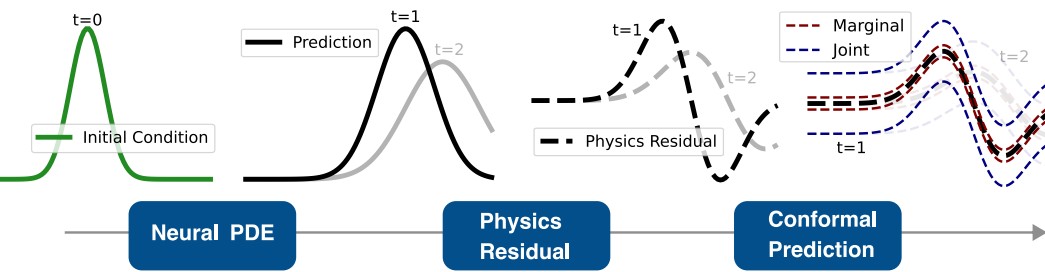

Figure 1: Schematic layout of the procedure for calibrated physics-informed uncertainty quantification. Initial conditions of the PDE are fed into a neural PDE to obtain predictions. PDE residuals are estimated over the predictions. The residual errors are calibrated using the marginal and joint formulation of conformal prediction to obtain statistically valid error bars. Marginal-CP leads to error bars at each point in space-time, whereas joint-CP provides error bars extending across the entire spatio-temporal domain. This figure is for illustrative purposes only.

## 2 RELATED WORK

Recently, CP, as a method of performing UQ, has been gaining popularity for usage with spatio-temporal data Sun (2022). Several works have explored the inductive CP framework for spatial and sequential data (Stankeviciute et al., 2021; Xu & Xie, 2021; Xu et al., 2023) including in the operator space (Ma et al., 2024). In Gopakumar et al. (2024a), the marginal-CP framework is extended to pre-trained as well as fine-tuned surrogate models for physical system modelling across an infinite-dimensional setting

The usage of PDE residuals under the guise of Physics-Informed Machine Learning (PIML) (Karniadakis et al., 2021) was made popular as an optimisation strategy for Physics-Informed Neural Networks (PINNs) (Raissi et al., 2019) and has found application in optimising neural operators (Li et al., 2024) and soft/hard enforcement of the physical constraints to deep learning models (Du et al., 2024; Chalapathi et al., 2024). However, they have rarely been used as a tool for providing UQ to the surrogate models; and where they have found application, UQ remained uncalibrated (Zhu et al., 2019). The majority of literature in UQ for neural PDE solvers has been looking at Bayesian methods, such as dropout, Bayesian neural networks and Monte Carlo methods (Geneva & Zabaras, 2020; Zou et al., 2024; Psaros et al., 2023), which lack guarantees or are computationally expensive.

## 3 BACKGROUND

### 3.1 NEURAL PDE SOLVERS

Consider the generic formulation of a PDE modelling the spatio-temporal evolution of $n$ field variables $u$ across a range of initial conditions:

$$D = D_t(u) + D_X(u) = 0 \qquad u \in \mathbb{R}^n,\ X \in \Omega,\ t \in [0, T], \qquad (1)$$
$$u(X, t) = g \qquad X \in \partial\Omega, \qquad (2)$$
$$u(X, 0) = a(\lambda, X). \qquad (3)$$

Here, $X$ defines the spatial domain bounded by $\Omega$, $t$ the temporal domain, $D_X$ and $D_t$, the composite operators of the associated spatial and temporal derivatives. The PDE is further defined by the boundary condition $g$ and initial condition $a$, which can be parameterised by $\lambda$. The set of solutions of field variables are expressed as $u \in \mathcal{U}$.

Neural PDE solvers aim to learn the behaviour governed by eq. (1) using a parameterised neural network ($\mathcal{NN}_\theta$). Starting from the initial conditions, the network is trained to solve the spatio-temporal evolution of the fields given by $\Omega \cup [0, T]$. Neural operators ($\mathcal{NO}_\theta$) are a special class of neural networks that learn the operator mapping from the function space of the PDE initial conditions $a \in \mathcal{A}$ to the function space of solutions $u \in \mathcal{U}$. A neural operator for solving an initial value

problem can be expressed as

$$\mathcal{U} = \mathcal{NO}_\theta(\mathcal{A}), \qquad u(X, t) = \mathcal{NO}_\theta\Big(u(X, 0)\Big) \qquad X \in \Omega, \ t \in [0, T]. \tag{4}$$

A **Fourier Neural Operator** (FNO) is an autoregressive neural operator that learns the spatio-temporal evolution of PDE solution by leveraging the Fourier transform as the kernel integrator (Li et al., 2021). The field evolution is learned using tuneable weight matrices of the network, parameterised directly in the Fourier space of the PDE solutions.

Since CP and our extension of it provide a post-hoc measure of quantifying the uncertainty of a neural PDE, it remains agnostic to model choice and training conditions. Considering the model independence of our approach, we restrict our experiments to modelling PDEs with an FNO. The FNO is chosen due to its cost-accuracy trade-off and efficiency as demonstrated by de Hoop et al. (2022) and Gopakumar et al. (2023b). CP over a range of neural-PDE solvers is has been applied by Gopakumar et al. (2024a), where the authors demonstrate that the coverage guarantees are upheld irrespective of the model choice, not needing us to experiment with various model architectures.

### 3.2 CONFORMAL PREDICTION

Conformal prediction (CP) (Shafer & Vovk, 2008; Vovk et al., 2005) is a statistical framework that addresses the accuracy of a predictive model. Consider a machine learning model $\hat{f} : \mathcal{X} \rightarrow \mathcal{Y}$ trained on a dataset $(X_i, Y_i)_{i=1}^N$, that is used to predict the next true label $Y_{n+1}$ at query point $X_{n+1}$. CP extends the point prediction $\tilde{Y}_{n+1}$ to a prediction set $\mathbb{C}^\alpha$, ensuring that

$$\mathbb{P}(Y_{n+1} \in \mathbb{C}^\alpha) \geq 1 - \alpha. \tag{5}$$

This guarantee, a function of the user-defined confidence level ($\alpha$ value), holds irrespective of the chosen model and training dataset. The only condition is that the calibration samples and the prediction samples are exchangeable. Traditional inductive CP partitions the labelled data into training and calibration sets (Papadopoulos, 2008). The performance of the model on the latter, measured using a *nonconformity score* is used to calibrate the model and obtain prediction sets.

Conventionally, nonconformity scores act on the model predictions and a labelled dataset (Kato et al., 2023). For deterministic models, they are often formulated as the Absolute Error Residual (AER) of the model predictions $\big(\hat{f}(X)\big)$ and targets $\big(Y\big)$. For probabilistic models, the score function (STD) is the absolute error of the prediction means $\big(\hat{f}_\mu(X)\big)$ and the targets $\big(Y\big)$, normalised by the standard deviation of the prediction

$$\hat{q}^\alpha = F_{\hat{s}}^{-1}\left( \frac{\lceil (n+1)(1-\alpha) \rceil}{n} \right). \tag{6}$$

$\big(\hat{f}_\sigma(X)\big)$. Having obtained a distribution of nonconformity scores ($\hat{s}$) of the calibration dataset $(X_i, Y_i)_{i=1}^n$, a quantile ($\hat{q}$) corresponding to the desired coverage ($1 - \alpha$) is estimated from its cumulative distribution function ($F_{\hat{s}}$) as given in eq. (6) (Papadopoulos, 2008). The quantile estimates the error bar associated with desired coverage and is combined with the new prediction to obtain the prediction sets. The nonconformity score functions and their prediction sets for AER and STD are given in eq. (7) and eq. (8), respectively.

## 4 PHYSICS RESIDUAL ERROR (PRE)

We introduce a novel **data-free** *nonconformity score* for Conformal Prediction (CP) to obtain statistically valid and guaranteed error bounds for neural PDE solvers. The Physics Residual Error (PRE) is defined as the PDE residual (Saad & Schultz, 1986) estimated over the discretised PDE solution obtained from the surrogate model. For an abstract PDE as in eq. (1), the PDE residual is the evaluation of the composite differential operator $D$. The PDE residual is treated as a score function by taking its L1 norm as indicated in eq. (9). While well-defined PDEs have solutions obeying eqs. (1) to (3), numerical solutions often fail to converge to the true solution (Pinder, 2018). Neural PDEs, trained on approximate numerical data, are further prone to non-convergent predictions. In PDE numerical analysis, the norm of the PDE residual is often used as a criterion for stability, convergence, and accuracy (Iserles, 2009). The PRE typically represents the violation of conservation laws

associated with the physical system. Using the residual error as a nonconformity score evaluates the neural PDE solver's performance by quantitatively estimating its effectiveness in modelling a PDE.

| Nonconformity | Score Function ($\mathcal{S}$) | Prediction Sets ($\mathcal{P}$) | |
|---|---|---|---|
| AER | $\hat{s} = \left( \left\| \hat{f}(X_i) - Y_i \right\| \right)_{i=1}^{n}$ | $\mathbb{C}^\alpha(X_{n+1}) = \hat{f}(X_{n+1}) \pm \hat{q}^\alpha$ | (7) |
| STD | $\hat{s} = \left( \dfrac{\left\| \hat{f}_\mu(X_i) - Y_i \right\|}{f_\sigma(X_i)} \right)_{i=1}^{n}$ | $\mathbb{C}^\alpha(X_{n+1}) = \hat{f}_\mu(X_{n+1}) \pm \hat{q}^\alpha \, \hat{f}_\sigma(X_{n+1})$ | (8) |
| PRE | $\hat{s} = \left( \left\| D(\hat{f}(X_i)) \right\| \right)_{i=1}^{n}$ | $\mathbb{C}^\alpha \left( D \big( \hat{f}(X_{n+1}) \big) \right) = \pm \hat{q}^\alpha$ | (9) |

The norm of the residual operator itself ($|D(\mathcal{NO}_\theta(u))|$) provides a measure of UQ for the neural PDE. However, it is limited by the accuracy of the gradient estimation method and can become computationally expensive when exploring a vast solution space (Tolsma & Barton, 1998). By using the residual norm as a nonconformity score, we are further calibrating the approximate physics residual error that is obtained by an inexpensive and coarse differential operator. CP using PRE provides statistically valid and guaranteed error bars across the PDE's residual space, incorporating physical information into the calibration procedure, and providing a calibrated measure of the physical misalignment of the surrogate model.

PRE as a nonconformity score enables **data-free conformal prediction**. The estimated scores rely only on the neural PDE predictions over a range of initial conditions, not on the target as in AER and STD. The only criterion is that the calibration and prediction domains arise from exchangeable initial conditions of the PDE. As shown in eq. (9), PRE allows for obtaining **"prediction sets" independent of the prediction inputs** (see appendix B for formalism). Traditional CP methods by design ensure that the predictions always fall within the estimated error bars (see eqs. (7) and (8)). They only provide guarantees that the true solution will obey coverage determined by $\alpha$. our CP-PRE formulation provides no guarantee that the true solution lies within the coverage bounds. Instead, it ensures that prediction outputs will lie within these bounds, allowing for the identification of predictions falling outside the coverage bounds $\mathbb{C}^\alpha$. Through our method, prediction sets can be validated without needing the true solution.

### 4.1 MARGINAL-CP

The CP formulation was initially conceptualised for calibrating univariate functions with single-point outputs (Vovk et al., 2005). It has recently been extended to spatio-temporal data, with multi-dimensional outputs with an immutable tensor structure (Gopakumar et al., 2024a). Within such spatio-temporal settings, CP has been implemented to provide marginal coverage, i.e. the calibration procedure provides independent error bars for each cell within the spatio-temporal domain. For an output tensor ($Y$) of shape ($Y \in \mathbb{R}^{N_x \times N_y \times N_t}$), where $N_x, N_y, N_t$ represent the spatio-temporal discretisation of the domain, marginal-CP uses the non-conformity scores outlined in eqs. (7) to (9) across each cell of $Y$ to obtain error bars which will be compliant eq. (5) for each cell. Marginal-CP using PRE helps indicate regions within a single prediction that lie outside the calibrated physical bounds and require specific attention, treating those predictions with caution.

### 4.2 JOINT-CP

The joint-CP formulation constructs a calibration procedure that provides coverage bands for multivariate functions. These coverage bands expand across the entire simulation domain $\Omega \times [0, T]$ (discretised as $\mathbb{R}^{N_x \times N_y \times N_t}$) rather than an individual cell within it. For a coverage band ($\mathbb{C}^\alpha$), the joint-CP formulation ensures that $1 - \alpha$ predictions/solutions would lie within the bounds. For performing joint-CP, the non-conformity scores are modified to reflect the supremum of the score functions $\mathcal{S}$ in eqs. (7) to (9). They are modulated by the standard-deviation $\sigma$ of the calibration scores (Diquigiovanni et al., 2021) to allow for obtaining prediction bands with varying widths

based on local behaviour (Diquigiovanni et al., 2022). The modifications of the score functions and prediction sets to perform CP are given by

$$\hat{s} = \sup_{X \in \Omega, \, t \in [0,T]} \left( \frac{\mathcal{S}}{\sigma(\mathcal{S})} \right), \tag{10}$$

$$\mathbb{C}^\alpha = \mathcal{P} \times \sigma(\mathcal{S}), \tag{11}$$

where $\mathcal{S}$ and $\mathcal{P}$ are the formulations of the nonconformity scores and prediction sets used for marginal-CP as shown in eqs. (7) to (9). Joint-CP becomes particularly useful in identifying predictions that fail to fall within coverage, allowing us to perform rejection sampling, where we can accept or reject a prediction based on a predetermined probability. Similar to that demonstrated by Casella et al. (2004), our framework can perform rejection sampling using a CP-based criterion. The acceptance probability is based on confidence level $\alpha$, allowing the joint-CP formulation using PRE to filter through predictions of the neural PDE solver. Upon being rejected, the initial conditions that led to those predictions could be provided to the expensive physics-based numerical PDE solver for further evaluation.

### 4.3 DIFFERENTIAL OPERATOR: FINITE-DIFFERENCE STENCILS AS CONVOLUTIONAL KERNELS

Calibrating neural PDEs using PRE nonconformity scores requires frequent evaluations of the composite differential operator ($D$) in eq. (1). For PDEs, this involves estimating numerous spatio-temporal gradients across the discretised domain, ranging from millions in simple cases to billions for complex physics. To address this computational challenge, we developed a scalable gradient estimation method for evaluating physics residual error.

We employ convolution operations with Finite Difference (FD) stencils as convolutional kernels for gradient estimation (Actor et al., 2020; Chen et al., 2024a;b). For instance, the 2D Laplacian operator $\nabla^2$, using a central difference scheme with discretisation $h$, can be approximated by

$$\nabla^2 \approx \frac{1}{h^2} \begin{bmatrix} 0 & 1 & 0 \\ 1 & -4 & 1 \\ 0 & 1 & 0 \end{bmatrix} \tag{12}$$

and used as a kernel. This approach is justified by the mathematical equivalence of FD approximations and discrete convolutions in the discretised domain. Both represent matrix-vector multiplications of a block Toeplitz matrix with a field vector (Strang, 1986; Fiorentino & Serra, 1991). The efficiency of this method stems from the optimised implementation of convolution operations in machine learning libraries like PyTorch (Paszke et al., 2019) and TensorFlow (Abadi et al., 2015). The Basic linear Algebra subroutines (BLAS) within these libraries leverage vectorisation and efficient memory access, resulting in significant performance improvements. Our experiments show a 1000x speed-up using *torch.nn.functional.conv3d* compared to a *numpy* implementation of the equivalent FD approximation on a standard CPU.

The FD approximation offers several advantages over Automatic Differentiation (AD) for our application. It is compatible with CP as a post-hoc measure, requires no architectural modifications, and is model-agnostic. Furthermore, FD implemented via convolutions is more memory-efficient than AD, which requires storing the entire computational graph. Our focus on the (mis)alignment of neural PDEs with eq. (1) allows us to disregard boundary conditions in our error bar estimations.

## 5 EXPERIMENTS

**Methodology:** Within each experiment, we start with data generation from a numerical solver, which is used to train a neural PDE solver, followed by UQ using CP. As laid out in fig. 1, the calibration procedure for demonstrating CP formulation with PRE involves three key steps: (a) sample model inputs from a bounded domain of initial conditions to generate predictions, (b) calculating PRE(s) for each prediction, and (c) using these PRE(s) as nonconformity scores to calibrate the model's physical error in residual space, employing both marginal and joint-CP formulations. Validation is performed by sampling initial conditions from the same bounds used in calibration. As shown in fig. 2, marginal-CP achieves guaranteed expected coverage, while joint-CP coverage fluctuates slightly around the guarantee due to its modulation function (see eqs. (10) and (11)).

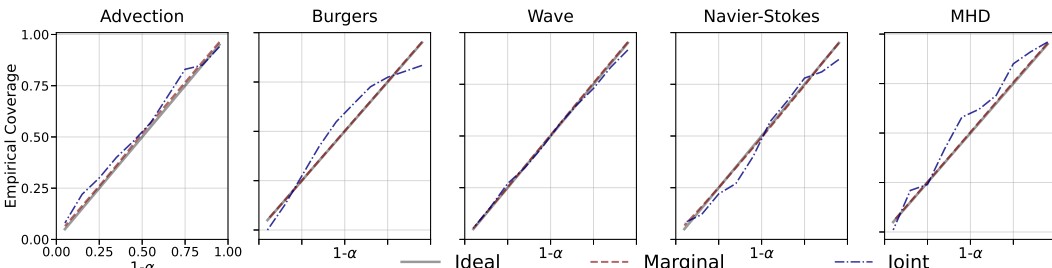

Figure 2: **Validation plots demonstrating coverage guarantee** detailed in eq. (5) obtained by performing CP using PRE across experiments. The average empirical coverage obtained by CP is given on the $y$-axis (ranging from 0 to 1, with 1 representing 100% coverage), while the coverage for which we calibrate is represented on the $x$-axis. We obtain guaranteed coverage while using marginal-CP formulation, and near-to-ideal coverage for the joint-CP formulation.

## 5.1 1D ADVECTION EQUATION

Consider the one-dimensional advection equation

$$\frac{\partial u}{\partial t} + v\frac{\partial u}{\partial x} = 0. \tag{13}$$

The state variable of interest $u$ is bounded within the domain $x \in [0,2], \ t \in [0,0.5]$ and moves within the domain at a constant velocity $v$. Data generation is performed by solving eq. (13) using a Crank-Nicolson method (Crank & Nicolson, 1947). Data is sampled using a parameterised initial condition that characterises the amplitude and position of the Gaussian field. Generated data is used to train a 1D FNO that takes in the initial condition and autoregressively with a step size of 1, learns to map the next 10 time frames. A reproducible script is attached in the supplementary material.

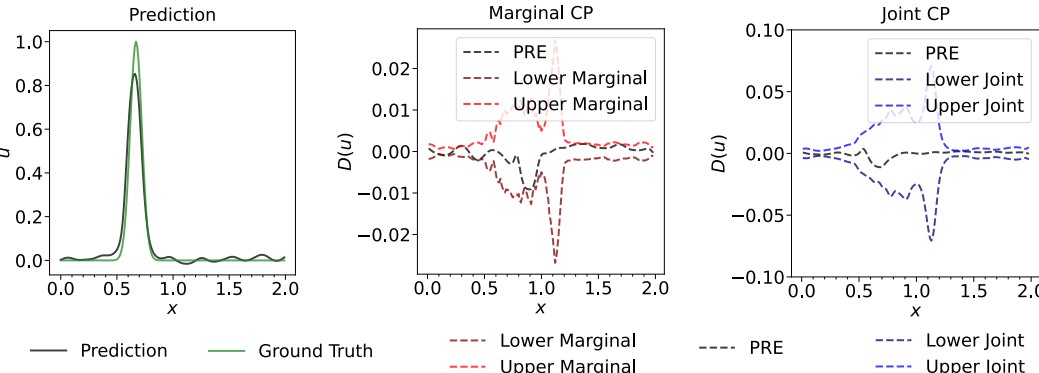

Figure 3: **Advection Equation:** (Left) Comparing the neural PDE (FNO) performance with that of the physics-based numerical solver at the last time instance. (Middle) Upper and lower bounds for 90% coverage obtained by performing marginal-CP. (Right) Upper and lower bounds for 90% coverage obtained by performing joint-CP. Marginal-CP provides tighter bounds for a prediction as opposed to joint-CP, whereas joint-CP provides a method of employing a relative sense of *reliability* of a prediction within a domain.

Figure 3 demonstrates the guaranteed bounds obtained over the residual space of eq. (13) utilising both the marginal and joint-CP formulations. Being cell-wise, marginal-CP guarantees coverage for each discretised point within the spatio-temporal domain. This allows for tighter bounds and error quantification of interested subdomains within regions but does not provide any guarantee across the entire prediction domain. Joint-CP acting across the entire domain provides a guarantee as to whether a prediction (instead of a single cell) will fall within the domain or not. Larger bounds are observed as they extend over the multivariate nature of the output. Though this comes with bigger bounds, it provides us with a mechanism to perform rejection-sampling of predictions. Within joint-CP, bounds dictating $1 - \alpha$ coverage suggest that approximately, $\alpha \times 100\%$ predictions from the

same domain will fall outside the bounds and can be rejected. Further details about the physics, parameterisation of the initial conditions, model and its training can be found in appendix E.

## 5.2 1D BURGERS EQUATION

Consider the 1D Burgers' Equation

$$\frac{\partial u}{\partial t} + u\frac{\partial u}{\partial x} = \nu\frac{\partial^2 u}{\partial x^2}. \tag{14}$$

The state variable of interest $u$ is bounded within the domain $x \in [0, 2], \ t \in [0, 1.25]$. The field is prescribed by a kinematic viscosity $\nu = 0.002$. Data is generated by solving eq. (14) using a spectral method (Canuto et al., 2007). Data sampled using a parameterised initial condition is used to train a 1D FNO that takes in the initial distribution of the state and learns to autoregressively predict the PDE evolution for the next 30 time frames.

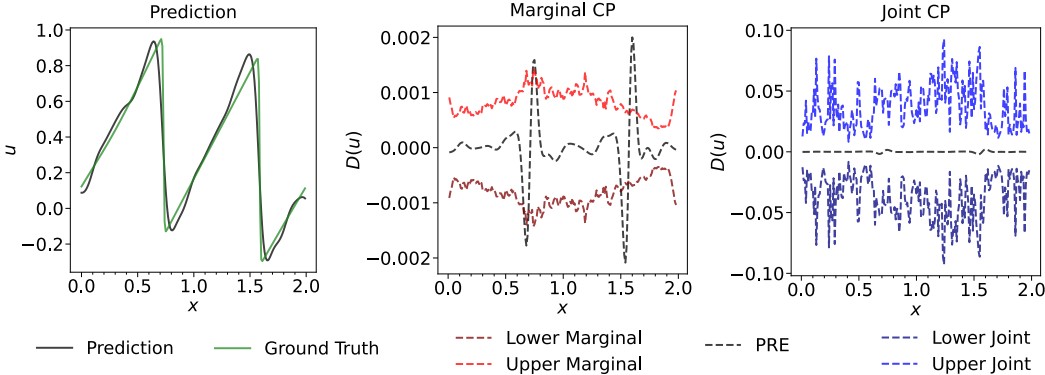

Figure 4: **Burgers' Equation:** (Left) Comparing the neural PDE (FNO) performance with that of the physics-based numerical solver at the last time instance. (Middle) Upper and lower bounds for 90% coverage obtained by performing marginal-CP. (Right) Upper and lower bounds for 90% coverage guaranteed by joint-CP over the residual space.

Figure 4 illustrates the guaranteed bounds over the residual space of eq. (14) using marginal and joint-CP formulations for 90% coverage. Marginal-CP provides cell-wise coverage, yielding tighter bounds for specific subdomains. Joint-CP provides bounds 50 times larger than that of the marginal-CP as it covers the entire prediction domain. Despite the large bounds, approximately $\alpha \times 100\%$ predictions fall outside it as given in fig. 2. For details on physics, initial condition parameterisation, model, and training, see appendix F.

## 5.3 2D WAVE EQUATION

Consider the two-dimensional wave equation

$$\frac{\partial^2 u}{\partial t^2} = c^2\left(\frac{\partial^2 u}{\partial x^2} + \frac{\partial^2 u}{\partial y^2}\right). \tag{15}$$

The state variable of interest $u$ is bounded within the domain $x \in [-1, 1], \ y \in [-1, 1], \ t \in [0, 1.0]$ and moves at a constant speed $c = 1.0$. Data is generated by solving eq. (15) using a spectral-based solver (Canuto et al., 2007) with periodic boundary conditions. Data is sampled using a parameterised initial condition of the amplitude and position of the Gaussian field. Generated data is used to train a 2D FNO starting from the initial condition, autoregressively learning the roll-out for the next 20 time frames.

Figure 5 provides a parallel comparison of the model prediction, the PRE over the prediction and the error bounds for 90% coverage obtained by performing marginal and joint-CP. Joint-CP creates bounds an order of magnitude higher than that of the marginal to account for the entire spatio-temporal domain. The utility of the PRE becomes clear in fig. 5, where for a seemingly well-fit



Figure 5: **Wave equation:** (From left to right) neural PDE (FNO) prediction at the last time instance, physics residual error of the prediction, Upper error bars obtained by performing marginal-CP and joint-CP respectively (90% coverage). For brevity, we have only shown the upper error bars of the symmetric prediction sets. $mod$ represents the modulation function in eq. (9).

prediction of the neural-pde the PRE is characterised by noisy features even in areas where the field should be zero. This is further visualised in fig. 16 in appendix G. By using CP, we are able to calibrate these features highlighting physical inconsistencies of the prediction. Our method also comes with the advantage of being data-free, which means that exchangeability becomes less of a concern as we can change the calibration domain and hence the prediction domain by simply reformulating the PRE accordingly. Further details about the experiment can be found in appendix G.

### 5.4 2D NAVIER-STOKES EQUATION

Consider the two-dimensional Navier-Stokes equations

$$\vec{\nabla} \cdot \vec{\mathbf{v}} = 0 \qquad \text{(Continuity equation)}, \qquad (16)$$

$$\frac{\partial \vec{\mathbf{v}}}{\partial t} + (\vec{\mathbf{v}} \cdot \vec{\nabla})\vec{\mathbf{v}} = \nu\nabla^2\vec{\mathbf{v}} - \nabla P \qquad \text{(Momentum equation)}, \qquad (17)$$

where we are interested in modelling the evolution of the velocity vector ($\vec{\mathbf{v}} = [u, v]$) and pressure ($P$) field of an incompressible fluid with kinematic viscosity ($\nu$). For data generation, eqs. (16) and (17) are solved on a domain $x \in [0,1]$, $y \in [0,1]$, $t \in [0,0.5]$ using a spectral-based solver (Canuto et al., 2007). A 2D multi-variable FNO (Gopakumar et al., 2024b) is trained to model the evolution of velocity and pressure autoregressively up until the $20^{th}$ time instance.

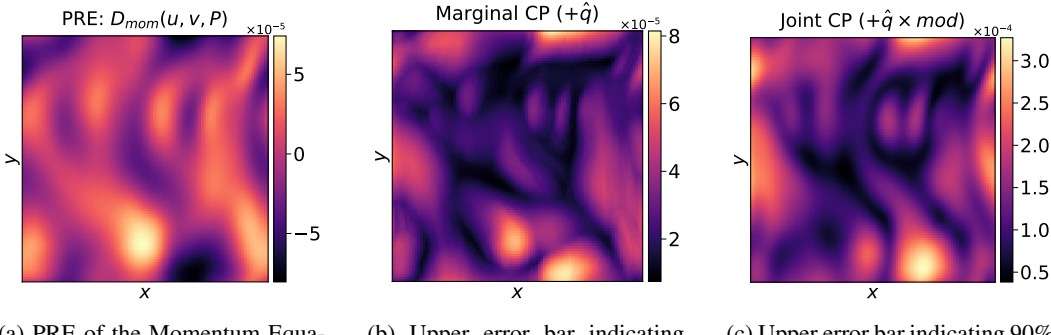

(a) PRE of the Momentum Equation over the FNO prediction.

(b) Upper error bar indicating 90% coverage with marginal-CP

(c) Upper error bar indicating 90% coverage with joint-CP

Figure 6: **Navier-Stokes:** CP using the Momentum Equation (17) as the PRE for a neural PDE surrogate model solving the Navier-Stokes equations.

Unlike previous examples, the Navier-Stokes case is comprised of two equations and hence has two PRE estimates: The continuity equation in eq. (16) and the momentum equation eq. (17), representing the conservation of mass and momentum respectively. Our method of performing CP over the residual space using PRE, allows us to calibrate the deviation of the model from the physical ground truth with respect to each equation. Figure 6 represents the PRE of the momentum equation over the FNO prediction, the upper bounds obtained by performing marginal and joint-CP over the FNO

prediction. In fig. 18, the same is depicted for the conservation of mass. Having two PDE residuals provides our framework added scrutiny in identifying relatively inconsistent predictions, as those that violate both bounds can be rejected easily. Further details about the physics, parameterisation of the initial conditions, model and its training can be found in appendix H.

## 5.5 2D MAGNETOHYDRODYNAMICS

Consider the magnetohydrodynamic (MHD) equations:

$$\frac{\partial \rho}{\partial t} + \vec{\nabla} \cdot (\rho \vec{v}) = 0 \qquad \text{(Continuity equation)}, \qquad (18)$$

$$\rho \left( \frac{\partial \vec{v}}{\partial t} + \vec{v} \cdot \nabla \vec{v} \right) = \frac{1}{\mu_0} \vec{B} \times (\vec{\nabla} \times \vec{B}) - \nabla P \qquad \text{(Momentum equation)}, \qquad (19)$$

$$\frac{\mathrm{d}}{\mathrm{d}t} \left( \frac{P}{\rho^{\gamma}} \right) = 0 \qquad \text{(Energy equation)}, \qquad (20)$$

$$\frac{\partial \vec{B}}{\partial t} = \vec{\nabla} \times (\vec{v} \times \vec{B}) \qquad \text{(Induction equation)}, \qquad (21)$$

$$\vec{\nabla} \cdot \vec{B} = 0 \qquad \text{(Gauß law for magnetism)}, \qquad (22)$$

where the density $(\rho)$, velocity vector $(\vec{v} = [u, v])$ and the pressure of plasma is modelled under a magnetic field $(\vec{B} = [B_x, B_y])$ across a spatio-temporal domain $x, y \in [0, 1]^2$, $t \in [0, 5]$. $\mu_0$ is the magnetic permeability of free space. Equations (18) to (22) represents the ideal MHD equations obtained as a combination of the Navier-Stokes equations for fluid flow with Maxwell's equations of electromagnetism (Alfvén, 1942; Gruber & Rappaz, 1985; Mocz et al., 2014). The equations assume perfect conductivity (no magnetic diffusivity) and no viscosity. We focus our experiment on the modelling of the Orszag-Tang vortex of a turbulent plasma (Orszag & Tang, 1979) with the data being generated using a finite volume method Eymard et al. (2000). A 2D FNO is trained to model the evolution of all 6 variables over a dataset generated by parameterised initial conditions.

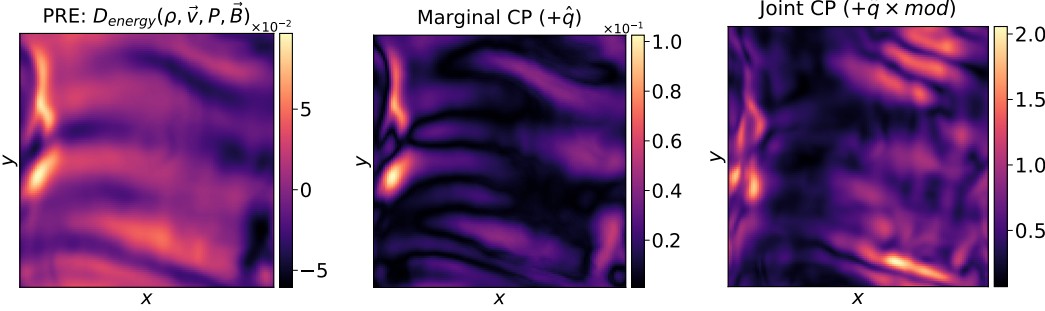

(a) PRE of the Energy Equation eq. (20) over the FNO prediction.

(b) Upper error bar indicating 90% coverage with marginal-CP

(c) Upper error bar indicating 90% coverage with joint-CP

Figure 7: **MHD:** CP using the Energy Equation (20) as the PRE for a neural PDE surrogate model solving the Ideal MHD equations. The last time instance of the prediction is shown.

Equations (18) to (22) provides us with 5 measures of estimating the PRE of the the MHD surrogate model. Each PRE estimate is dependent on a different set of variables associated with the system and allows us to infer errors contributed to each variable accordingly. In fig. 7 CP is performed using the energy equation eq. (20), which depends on all of the 6 variables and has a higher error value than that of the induction equation as shown in fig. 8 which is dependent on the velocity and magnetic fields only. Plots indicating CP utilising the other residuals (figs. 20 and 21) as well as further details about the physics and the surrogate model can be found in appendix I.

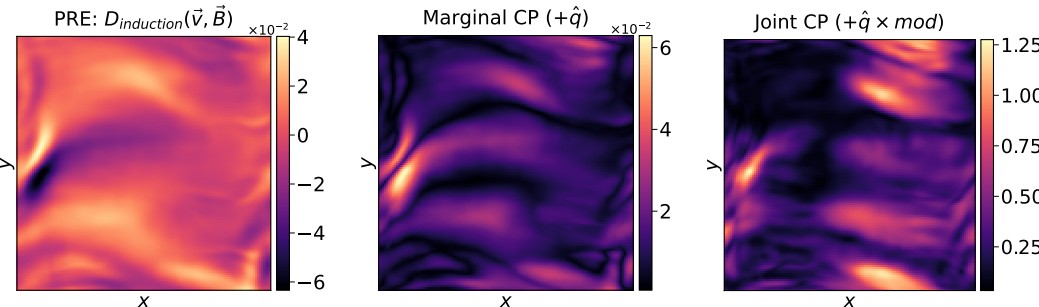

(a) PRE of the Induction Equation eq. (21) over the FNO prediction.

(b) Upper error bar indicating 90% coverage with marginal-CP.

(c) Upper error bar indicating 90% coverage with joint-CP

Figure 8: **MHD:** CP using the Induction eq. (21) as the PRE for a neural PDE surrogate model solving the Ideal MHD equations. The last time instance of the prediction is shown.

## 6 DISCUSSION

If "All models are wrong, but some are useful" (Box, 1976), through this work, we explore a novel framework for providing data-free, model and domain agnostic measure of *usefulness* of neural PDE models. We deploy a principled method of evaluating the accuracy of the solution, i.e. its (calibrated) obedience to the known physics of the system under study. As opposed to other methods of UQ for neural PDEs, our method is physics-informed and offers coverage guarantees. This calibration procedure is not limited to PDE modelling but may apply to ODEs or any other scenario where the model outputs can be framed as a residual. We conclude with a discussion of the strengths, limitations and potential improvements.

**Strengths**   The PRE estimates the violation of conservation laws in neural PDE predictions, enabling calibration of physics deviation with statistical coverage guarantees. This post-hoc uncertainty quantification is model- and physics-agnostic, scaling linearly with model complexity and quasi-linearly with PDE complexity due to the additive nature of differential operators. Our framework reformulates CP to be data-free, expressing model inaccuracy solely through PRE, and does not require a labelled dataset. This approach reduces calibration costs and loosens exchangeability restrictions. The PRE formulation (section 4, appendix B) yields input-independent prediction sets, allowing for the identification of weak predictions within single simulations (marginal-CP) and across multiple predictions (joint-CP). The latter enables rejection sampling, potentially serving as an active-learning pipeline for neural PDE solvers (Musekamp et al., 2024).

**Limitations**   Our method's coverage bounds exist in the PDE residual space rather than the Euclidean space of physical variables. Transforming to physical space involves challenging set propagation through integral operations, which may require linear approximations (Teng et al., 2023) or expensive MCMC sampling (Andrieu et al., 2003) for complex PDEs. The data-free approach lacks a grounding target for calibration, though we argue that a large sample of model outputs provides a statistically significant overview of uncertainty. The sampling cost from the neural-PDE solver for calibration must be considered. The modulation function in Equation (11) improves local bound representation but compromises CP guarantee assumptions in the joint setting (fig. 2). PRE estimation using finite-difference stencils also introduces the errors associated with Taylor expansion. The current formulation is limited to regular grids with fixed spacing, though extensions to unstructured grids via graph convolutions are possible (Eliasof & Treister, 2020).

## 7 CONCLUSION

The PRE-CP formulation provides a novel framework for providing guaranteed and physics-informed uncertainty estimates for each cell within a prediction as well as across predictions. Our work enhances the reliability of neural PDE solvers, potentially broadening their applicability in science and engineering domains where robust uncertainty quantification is crucial.

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

## A    THEOREM: DATA-FREE CP

**Preliminaries:** Let $D : \mathbb{R}^m \to \mathbb{R}^m$ be a physics residual operator mapping a function to its PDE residual value, where: $\{X_i\}_{i=1}^n$ is the calibration set, $\hat{f}$ is the model, $\hat{q}^\alpha$ is estimated as the $\lceil (n+1)(1-\alpha) \rceil / n$ -quantile of $\{|D(\hat{f}(X_i))|\}_{i=1}^n$

**Theorem 1.** *If the residuals $\{D(\hat{f}(X_i))\}_{i=1}^{n+1}$ are exchangeable random variables, then for any significance level $\alpha \in (0,1)$ and any new input $X_{n+1}$ we have the following coverage guarantee:*

$$\mathbb{P}(|\mathcal{D}(\hat{f}(X_{n+1}))| \in C_\alpha) \geq 1 - \alpha \, ; \;\; C_\alpha = [-\hat{q}_\alpha, \hat{q}_\alpha]$$

*Proof.* Let $R_i = |D(\hat{f}(X_i))|$ for $i = 1, \ldots, n+1$. We have, by assumption, $(R_1, \ldots, R_n, R_{n+1})$ is an exchangeable sequence. Define the rank $\pi$ of $R_{n+1}$ w.r.t. all other residuals:

$$\pi(R_{n+1}) = |\{i = 1, \ldots, n+1 : R_i \leq R_{n+1}\}|$$

By exchangeability, the rank $\pi(R_{n+1})$ is uniformly distributed over $\{1, \ldots, n+1\}$. Therefore,

$$P(\pi(R_{n+1}) \leq \lceil (n+1)(1-\alpha) \rceil) = \frac{\lceil (n+1)(1-\alpha) \rceil}{n} \geq 1 - \alpha.$$

By construction of $\hat{q}^\alpha$ we have that,

$$\{\pi(R_{n+1}) \leq \lceil (n+1)(1-\alpha) \rceil\} \subseteq \{R_{n+1} \leq \hat{q}^\alpha\}.$$

Putting this together,

$$P(|D(\hat{f}(X_{n+1}))| \leq \hat{q}^\alpha) = P(R_{n+1} \leq \hat{q}^\alpha) \geq 1 - \alpha,$$

which completes the proof. $\qquad\qquad\qquad\qquad\qquad\qquad\qquad\qquad\qquad\qquad\qquad\qquad\qquad\quad\square$

## B   PRE: SCORE FUNCTION AND PREDICTION SETS

For a general nonconformity score $S$, the prediction set for a new input $X_{n+1}$ is typically defined as:

$$\mathbb{C}^\alpha(X_{n+1}) = \{y : S(X_{n+1}, y) \leq \hat{q}^\alpha\},$$

where $\hat{q}^\alpha$ is the $(1 - \alpha)$-quantile of the nonconformity scores on the calibration set.

For AER and STD, the nonconformity scores depend on both the input $X$ and the output (target) $Y$:

$$S_{AER}(X, Y) = |\hat{f}(X) - Y|,$$

$$S_{STD}(X, Y) = \frac{|\hat{f}_\mu(X) - Y|}{\hat{f}_\sigma(X)}.$$

The resulting prediction sets are:

$$\mathbb{C}^\alpha_{AER}(X_{n+1}) = [\hat{f}(X_{n+1}) - \hat{q}^\alpha, \hat{f}(X_{n+1}) + \hat{q}^\alpha],$$

$$\mathbb{C}^\alpha_{STD}(X_{n+1}) = [\hat{f}_\mu(X_{n+1}) - \hat{q}^\alpha \hat{f}_\sigma(X_{n+1}), \hat{f}_\mu(X_{n+1}) + \hat{q}^\alpha \hat{f}_\sigma(X_{n+1})].$$

These prediction sets clearly depend on the input $X_{n+1}$.

For PRE, the nonconformity score depends only on the model output and not on the target:

$$S_{PRE}(\hat{f}(X)) = |D(\hat{f}(X)) - 0|,$$

where $D$ is the PDE residual operator. The key difference is that the true output $Y$ for PRE, irrespective of the PDE is always 0 and does not depend on the input $X$. PRE is a measure of how well the model output satisfies the physics rather than how it fits certain data. Hence, we can formulate a nonconformity score that is data-free and eventually leads to input-independent prediction sets as given below.

For PRE, we can reframe the prediction set definition:

$$\mathbb{C}^\alpha_{PRE} = \{\hat{f}(X) : |D(\hat{f}(X))| \leq \hat{q}^\alpha\}.$$

This set is not defined in terms of the true $Y$ values but in terms of the allowable model outputs $\hat{f}(X)$ that satisfy the PDE residual constraint. Thus, the prediction set can be expressed as:

$$\mathbb{C}^\alpha_{PRE} = [-\hat{q}^\alpha, \hat{q}^\alpha].$$

This formulation is independent of the input $X$, as it only depends on the quantile $\hat{q}^\alpha$ derived from the calibration set as given in eq. (6).

To validate predictions using PRE:

1. For a new input $X_{n+1}$, compute $\hat{f}(X_{n+1})$.
2. Calculate the residual: $r = |D(\hat{f}(X_{n+1}))|$.
3. Check if $r \in [-\hat{q}^\alpha, \hat{q}^\alpha]$ for a given $\alpha$.

If the condition in step 3 is satisfied, the error bounds dictated by $[-\hat{q}^\alpha, \hat{q}^\alpha]$ is considered valid according to the CP framework, regardless of the specific input $X_{n+1}$.

# C   COMPARISON TO OTHER UQ METHODS

| Method | Data-Free | Modification-Free | Sampling-Free | Guaranteed Coverage | Physics-Informed |
|---|---|---|---|---|---|
| MC Dropout | ✓ | ✗ | ✗ | ✗ | ✗ |
| Deep Ensemble | ✓ | ✗ | ✗ | ✗ | ✗ |
| BNN | ✓ | ✗ | ✗ | ✗ | ✗ |
| SWA-G | ✓ | ✗ | ✗ | ✗ | ✗ |
| CP-AER | ✗ | ✓ | ✓ | ✓ | ✗ |
| CP-PRE (Ours) | ✓ | ✓ | ✓ | ✓ | ✓ |

Table 1: Comparing features across various UQ measures. Our method is data-free, does not require any modifications or sampling, and helps obtain guaranteed coverage bounds in a physics-informed manner.

Table 2: Wave Equation - Coverage measured for $2\sigma (\sim 95\%)$

| | in-distribution | | out-distribution | | Time | |
|---|---|---|---|---|---|---|
| UQ | L2 | Coverage | L2 | Coverage | Train (hr) | Eval (s) |
| Deterministic | 1.77e-05 ± 3.69e-07 | - | 2.46e-03 ± 2.00e-05 | - | 0:38 | 22 |
| MC Dropout | 1.44e-04 ± 3.26e-06 | 97.31 ± 0.03 | 2.12e-03 ± 2.60e-05 | 89.83 ± 0.07 | 0:52 | 120 |
| Deep Ensemble | 8.76e-06 ± 2.43e-07 | 98.02 ± 0.04 | 2.42e-03 ± 1.58e-05 | 83.44 ± 0.12 | 3:10 | 112 |
| BNN | 1.92e-04 ± 1.92e-06 | 97.10 ± 0.09 | 2.67e-03 ± 1.26e-05 | 91.76 ± 0.10 | 0:53 | 118 |
| SWA-G | 1.41e-05 ± 1.74e-06 | 94.55 ± 3.25 | 2.55e-03 ± 2.82e-05 | 81.90 ± 3.31 | 0:47 | 113 |
| CP-AER | 1.76e-05 ± 4.40e-07 | 95.70 ± 0.21 | 2.46e-03 ± 1.41e-05 | 95.59 ± 0.14 | 0:38 | 23 |
| CP-PRE (Ours) | 1.78e-05 ± 4.61e-07 | 95.52 ± 0.21 | 2.46e-03 ± 1.25e-05 | 95.39 ± 0.12 | 0:38 | 23 |

Table 3: Navier-Stokes Equations - Coverage measured for $2\sigma (\sim 95\%)$

| | in-distribution | | out-distribution | | Time | |
|---|---|---|---|---|---|---|
| UQ | L2 | Coverage | L2 | Coverage | Train (hr) | Eval (s) |
| Deterministic | 1.05e-04 ± 6.91e-06 | - | 3.67e-03 ± 5.30e-05 | - | 3:22 | 25 |
| MC Dropout | 5.96e-04 ± 2.30e-05 | 82.21 ± 0.22 | 4.30e-03 ± 8.05e-05 | 44.05 ± 0.26 | 3:34 | 153 |
| Deep Ensemble | 1.22e-04 ± 3.95e-06 | 91.31 ± 0.08 | 3.67e-03 ± 3.52e-05 | 30.74 ± 0.19 | 16:22 | 147 |
| BNN | 6.90e-03 ± 1.31e-04 | 89.91 ± 0.20 | 6.95e-03 ± 1.31e-04 | 85.19 ± 0.23 | 3:39 | 152 |
| SWA-G | 1.96e-04 ± 1.15e-05 | 84.22 ± 2.37 | 3.63e-03 ± 1.37e-04 | 31.00 ± 2.85 | 3:28 | 146 |
| CP-AER | 1.05e-04 ± 6.58e-06 | 95.56 ± 0.40 | 3.66e-03 ± 2.81e-05 | 95.54 ± 0.15 | 3:22 | 26 |
| CP-PRE (Ours) | 1.07e-04 ± 5.18e-06 | 95.44 ± 0.22 | 3.70e-03 ± 4.23e-05 | 95.57 ± 0.14 | 3:22 | 34 |

Table 4: Magnetohydrodynamic Equations - Coverage measured for $2\sigma (\sim 95\%)$

| | in-distribution | | out-distribution | | Time | |
|---|---|---|---|---|---|---|
| UQ | L2 | Coverage | L2 | Coverage | Train (hr) | Eval (s) |
| Deterministic | 2.20e-03 ± 5.20e-03 | - | 4.71e-02 ± 1.06e-03 | - | 5:00 | 40 |
| MC Dropout | 3.29e-02 ± 5.86e-04 | 41.13 ± 0.19 | 2.09e-01 ± 1.38e-03 | 16.91 ± 0.06 | 5:30 | 240 |
| Deep Ensemble | 3.59e-03 ± 3.51e-04 | 78.15 ± 0.16 | 3.41e-01 ± 3.15e-02 | 39.63 ± 0.31 | 26:25 | 235 |
| BNN | 4.20e-03 ± 4.08e-05 | 90.24 ± 0.10 | 4.63e-02 ± 8.98e-04 | 62.37 ± 0.46 | 5:40 | 240 |
| SWA-G | 2.61e-03 ± 9.68e-05 | 48.50 ± 3.81 | 4.53e-02 ± 6.64e-04 | 14.22 ± 1.35 | 5:22 | 236 |
| CP-AER | 2.20e-03 ± 4.38e-05 | 95.61 ± 0.26 | 4.69e-02 ± 8.18e-04 | 95.60 ± 0.27 | 5:00 | 42 |
| CP-PRE (Ours) | 2.20e-03 ± 4.96e-03 | 95.54 ± 0.18 | 4.71e-02 ± 1.06e-03 | 95.67 ± 0.22 | 5:00 | 82 |

TABLE INDEX

**Deterministic**: Vanilla FNO (Li et al., 2021)

**MC Dropout**: FNO with Dropout (Gal & Ghahramani, 2016)

**Deep Ensemble**: Ensemble of FNOs (Lakshminarayanan et al., 2017)

**SWA-G**: Stochastic Weighted Averaging - Gaussian (Maddox et al., 2019)

**in-distribution**: Model evaluated on initial states sampled from the same parameter range (as given in the appendix) of the initial condition as used in the training data.

**out-distribution**: Model evaluated on initial states sampled from a different parameter range of the initial conditions as used in the training data.

**L2**: L2 norm of the model output with the ground truth in the normalised domain.

**Coverage**: Percentage coverage of the model outputs within the estimated error bounds

**Train Time**: Training time on a single A100 GPU.

**Eval. Time**: Evaluation time on a single A100 GPU.

# D CONVOPERATOR: CONVOLUTIONAL KERNELS FOR GRADIENT ESTIMATION

Within the code base for this paper, we release a utility function that constructs convolutional layers for gradient estimation based on your choice of order of differentiation and Taylor approximation. This allows for the PRE score function to be easily expressed in a single line of code [1]

This section provides an overview of the code implementation and algorithm for estimating the PRE using Convolution operations. We'll use an arbitrary PDE example with a temporal gradient $\frac{\partial u}{\partial t}$ and a Laplacian $\left( \frac{\partial^2}{\partial x^2} + \frac{\partial^2}{\partial y^2} \right)$ to illustrate the process.

$$\frac{\partial u}{\partial t} - \alpha \left( \frac{\partial^2 u}{\partial x^2} + \frac{\partial^2 u}{\partial y^2} \right) + \beta u = 0, \tag{23}$$

where $u$ is the field variable, $t$ is time, $x$ and $y$ are spatial coordinates, and $\alpha$ and $\beta$ are constants. To estimate the PDE residual given by eq. (23), we need to estimate the associated spatio-temporal gradients.

First, we use the `ConvOperator` class from `Utils/ConvOps_2d.py` to set up the convolutional layer with kernels taken from the appropriate finite difference stencils:

```
from ConvOps_2d import ConvOperator

# Define each operator within the PDE
D_t = ConvOperator(domain='t', order=1) #time-derivative
D_xx_yy = ConvOperator(domain=('x','y'), order=2) #Laplacian
D_identity = ConvOperator() #Identity Operator
```

The `ConvOperator` class is used to set up a gradient operation. It takes in `variable(s) of differentiation` and `order of differentiation` as arguments to design the appropriate forward difference stencil and then sets up a convolutional layer with the stencil as the kernel. Under the hood, the class will take care of devising a 3D convolutional layer, and setup the kernel so that it acts on a spatio-temporal tensor of dimensionality: *[BS, Nt, Nx, Ny]* which expands to batch size, temporal discretisation and the spatial discretisation in $x$ and $y$.

```
alpha, beta = 1.0, 0.5  # Example coefficients
D = ConvOperator() #Additive Kernels
D.kernel = D_t.kernel - alpha * D_xx_yy.kernel - beta * D_identity.kernel
```

The convolutional kernels are additive i.e. in order to estimate the residual in one convolutional operation, they could be added together to form a composite kernel that characterises the entire PDE residual.

Once having set up the kernels, PRE estimation is as simple as passing the composite class instance $D$ the predictions from the neural PDE surroga te (ensuring that the output is in the same order as the kernel outlined above).

```
y_pred = model(X)
PRE = D(y_pred)
```

Only operating on the outputs, this method of PRE estimation is memory efficient, computationally cheap and with the `ConvOperator` evaluating the PDE residual can be done in a single line of code.

## D.1 IMPACT OF DISCRETISATION

As demonstrated in (Bartolucci et al., 2023), the discretisation of the inputs and hence model outputs plays an important role in the accuracy of the neural-PDE solvers. Though the neural operators are constructed for discretisation-invariant behaviour due to the band-limited nature of the functions,

---

[1]The code and associated utility functions can be found in this github repository.

they often exhibit discretisation-convergent behaviour rather than be fully discretisation-invariant. This is of particular importance in the temporal dimensions as these neural-PDE models utilise a discrete, autoregressive based time-stepping and is baked into the model within its training regime (McCabe et al., 2023). Due to lack of control in the discretisation within the temporal domain $(dt)$, the PRE estimates tend to have higher numerical errors as well. In fig. 9, we visualise the evaluation of finite difference in 2D+time as a 3D convolution. The finite difference stencil i.e. the convolutional kernel has a unit discretisation of $dx$, $dy$ and $dt$ associated with the problem and is applied over the signal i.e. the output from the neural-PDE $u$ spanning the domain $x, y, t$, where $x \in [0, X]$, $y \in [0, Y]$, $t \in [0, T]$.

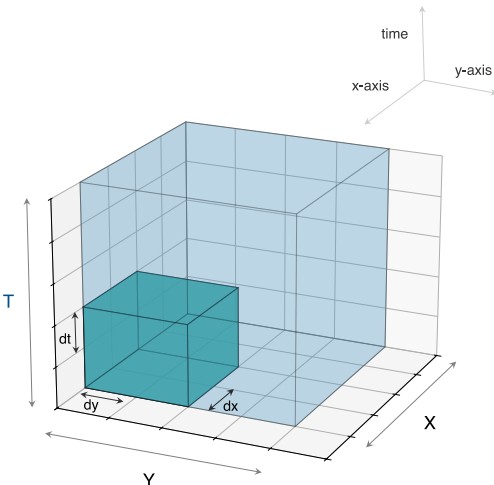

Figure 9: PRE estimation using the 3D convolutions with finite difference stencils as convolutional kernels being applied over the neural-PDE predictions as the signals.

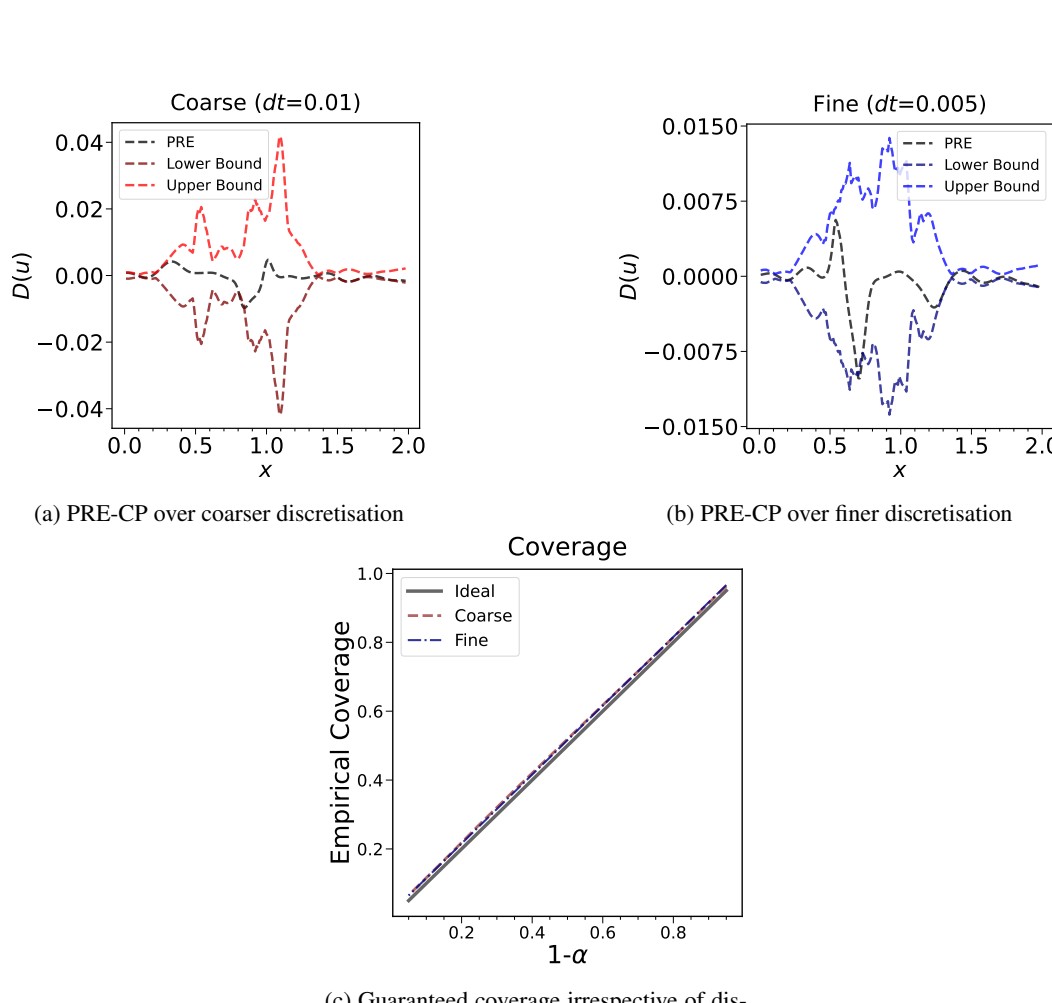

(a) PRE-CP over coarser discretisation

(b) PRE-CP over finer discretisation

(c) Guaranteed coverage irrespective of discretisation error.

Figure 10: PRE-CP provides guaranteed coverage irrespective of the discrertisation associated with the model outputs., however, the width of the obtained coverage bounds indicates the discretisation error associated with the gradient estimation. Coverage taken for $\alpha = 0.1 \sim 90\%$ coverage.

# E   1D ADVECTION EQUATION

## E.1   PHYSICS

Consider the one-dimensional advection equation, parameterised by the initial condition:

$$\frac{\partial u}{\partial t} = vD\frac{\partial u}{\partial x}, \quad x \in [0,2], \ t \in [0,0.5],$$
$$u(x, t = 0) = Ae^{(x-X)^2}. \tag{24}$$

Here $u$ defines the density of the fluid, $x$ the spatial coordinate, $t$ the temporal coordinate and $v$ the advection speed. initial condition is parameterised by $A$ and $X$, representing the amplitude and position of a Gaussian distribution. A no-flux boundary condition bounds the system.

The numerical solution for the above equation is built using a finite difference solver with a crank-nicolson method implemented in Python. We construct a dataset by performing a Latin hypercube sampling across parameters $A, X$. Each parameter is sampled from within the domain given in table 5 to generate 100 simulation points, each with its own initial condition. Each simulation is run for 50-time iterations with a $\Delta t = 0.01$ across a spatial domain spanning [0,2], uniformly discretised into 200 spatial units in the x-axis.

Table 5: Domain range of initial condition parameters for the 1D advection equation.

| Parameter | Domain | Type |
|---|---|---|
| Amplitude ($A$) | $[50, 200]$ | Continuous |
| Position ($X$) | $[0.5, 1.0]$ | Continuous |

## E.2   MODEL AND TRAINING

We use a one-dimensional FNO to model the evolution of the convection-diffusion equation. The FNO learns to perform the mapping from the initial condition to the next time instance, having a step size of 1. The model autoregressively learns the evolution of the field up until the $10^{th}$ time instance. Each Fourier layer has 8 modes and a width of 16. The FNO architecture can be found in table 6. Considering the field values governing the evolution of the advection equation are relatively small, we avoid normalisations. The model is trained for up to 100 epochs using the Adam optimiser Kingma & Ba (2015) with a step-decaying learning rate. The learning rate is initially set to 0.005 and scheduled to decrease by half after every 100 epochs. The model was trained using an LP-loss (Gopakumar et al., 2024b).

Table 6: Architecture of the 1D FNO deployed for modelling 1D Advection Equation

| Part | Layer | Output Shape |
|---|---|---|
| Input | - | (50, 1, 200, 1) |
| Lifting | Linear | (50, 1, 200, 16) |
| Fourier 1 | Fourier1d/Conv1d/Add/GELU | (50, 1, 16, 200) |
| Fourier 2 | Fourier1d/Conv1d/Add/GELU | (50, 1, 16, 200) |
| Fourier 3 | Fourier1d/Conv1d/Add/GELU | (50, 1, 16, 200) |
| Fourier 4 | Fourier1d/Conv1d/Add/GELU | (50, 1, 16, 200) |
| Fourier 5 | Fourier1d/Conv1d/Add/GELU | (50, 1, 16, 200) |
| Fourier 6 | Fourier1d/Conv1d/Add/GELU | (50, 1, 16, 200) |
| Projection 1 | Linear | (50, 1, 200, 256) |
| Projection 2 | Linear | (50, 1, 200, 1) |

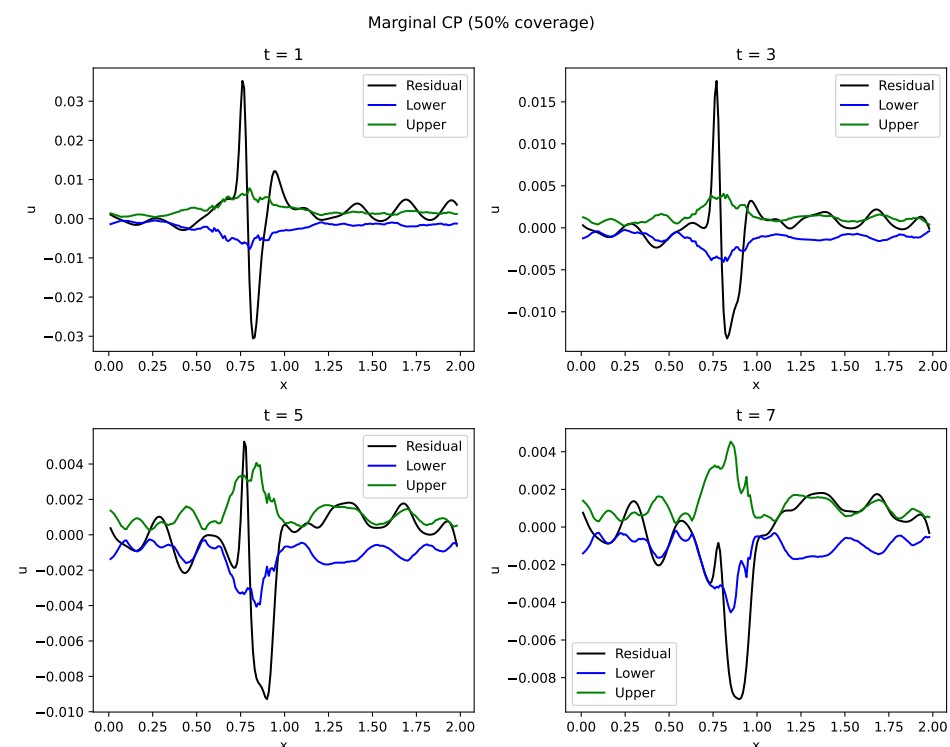

Figure 11: Advection Equation: Marginal-CP with $\alpha = 0.5$

### E.3 CALIBRATION AND VALIDATION

To perform the calibration as outlined in section 5, model predictions are obtained using initial conditions sampled from the domain given in table 5. The same bounded domain for the initial condition parameters is used for calibration and validation. 100 initial conditions are sampled and fed to the model to obtain and prediction for both the calibration and the validation.

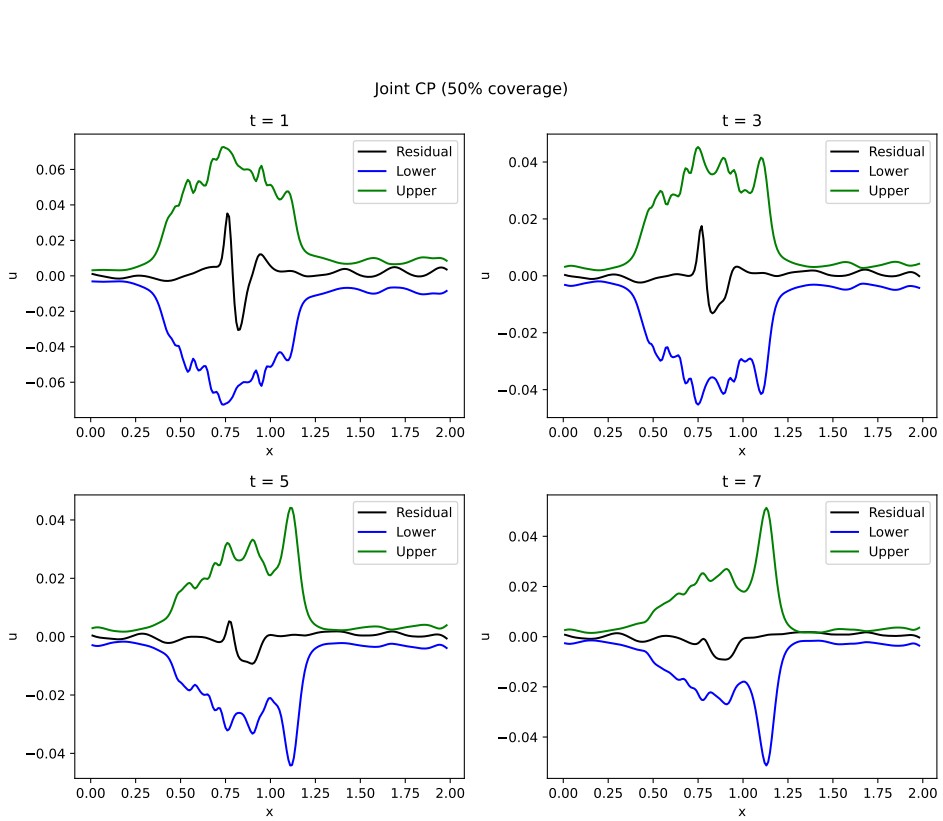

Figure 12: Advection Equation: joint-CP with $\alpha = 0.5$

# F 1D BURGERS EQUATION

## F.1 PHYSICS

Consider the one-dimensional Burgers' equation:

$$\frac{\partial u}{\partial t} + u\frac{\partial u}{\partial x} = \nu\frac{\partial^2 u}{\partial x^2},$$

$$u(x, t = 0) = \sin(\alpha\pi x) + \cos(-\beta\pi x) + \frac{1}{\cosh(\gamma\pi x)}, \tag{25}$$

where $u$ defines the field variable, $\nu$ the kinematic viscosity, $x$ the spatial coordinate, $t$ the temporal coordinates. $\alpha, \beta$ and $\gamma$ are variables that parameterise the initial condition of the PDE setup. The system is bounded periodically within the mentioned domain.

The solution for the Burgers' equation is obtained by deploying a spectral solver (Canuto et al., 2007). The dataset is built by performing a Latin hypercube scan across the defined domain for the parameters $\alpha, \beta, \gamma$, sampled for each simulation. We generate 1000 simulation points, each one with its initial condition and use it for training.

The physics of the equation, given by the various coefficients is held constant across the dataset generation throughout as given in eq. (25). Each data point, as in each simulation is generated with a different initial condition as described above. The parameters of the initial conditions are sampled from within the domain as given in table 7. Each simulation is run for 500-time iterations with a $\Delta t = 0.0025$ across a spatial domain spanning $[0, 2]$, uniformly discretised into 1000 spatial units in the x and y axes. The temporal domain is subsampled to factor in every $10^{th}$ time instance, while the spatial domain is downsampled to every $5^{th}$ instance.

Table 7: Domain range of initial condition parameters for the 1D Burgers' equation.

| Parameter | Domain | Type |
|---|---|---|
| $\alpha$ | $[-3, 3]$ | Continuous |
| $\beta$ | $[-3, 3]$ | Continuous |
| $\gamma$ | $[-3, 3]$ | Continuous |

## F.2 MODEL AND TRAINING

We train a 1D FNO to map the spatio-temporal evolution of the field variables. We deploy an auto-regressive structure that performs time rollouts allowing us to map the initial distribution recursively up until the $30^{th}$ time instance with a step size of 1. Each Fourier layer has 8 modes and a width of 32. The FNO architecture can be found in table 8. We employ a linear range normalisation scheme, placing the field values between -1 and 1. Each model is trained for up to 500 epochs using the Adam optimiser (Kingma & Ba, 2015) with a step-decaying learning rate. The learning rate is initially set to 0.005 and scheduled to decrease by half after every 100 epochs. The model was trained using an LP-loss (Gopakumar et al., 2024b).

## F.3 CALIBRATION AND VALIDATION

To perform the calibration as outlined in section 5, model predictions are obtained using initial conditions sampled from the domain given in table 7. The same bounded domain for the initial condition parameters is used for calibration and validation. 1000 initial conditions are sampled and fed to the model to perform the calibration and 100 samples are gathered for performing the validation.

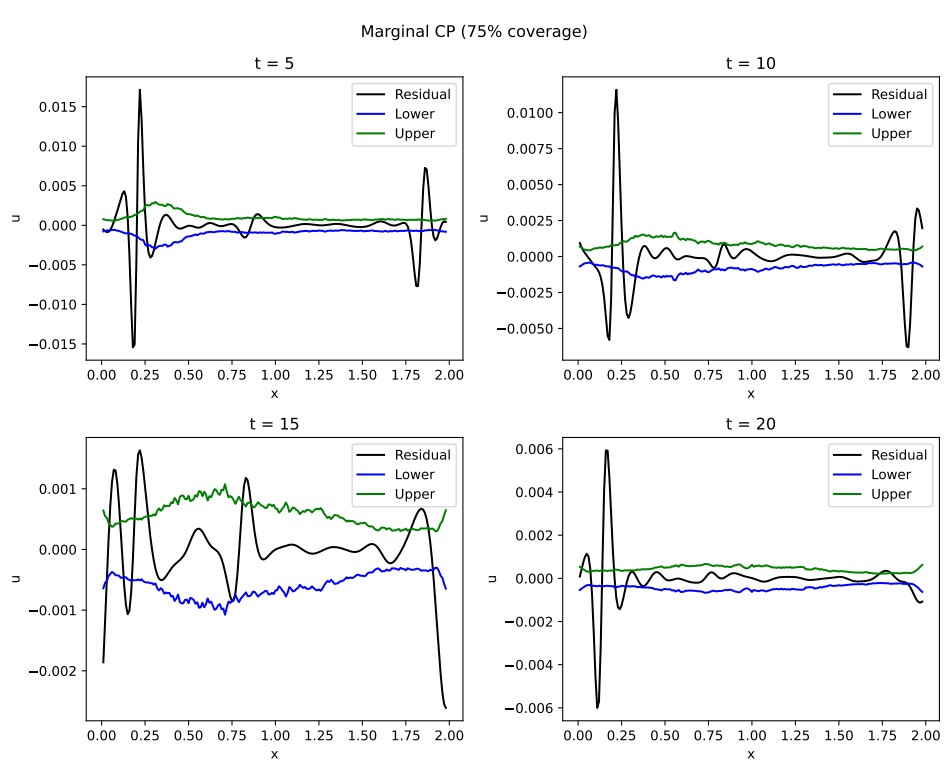

Figure 13: Burgers Equation: Marginal-CP with $\alpha = 0.75$

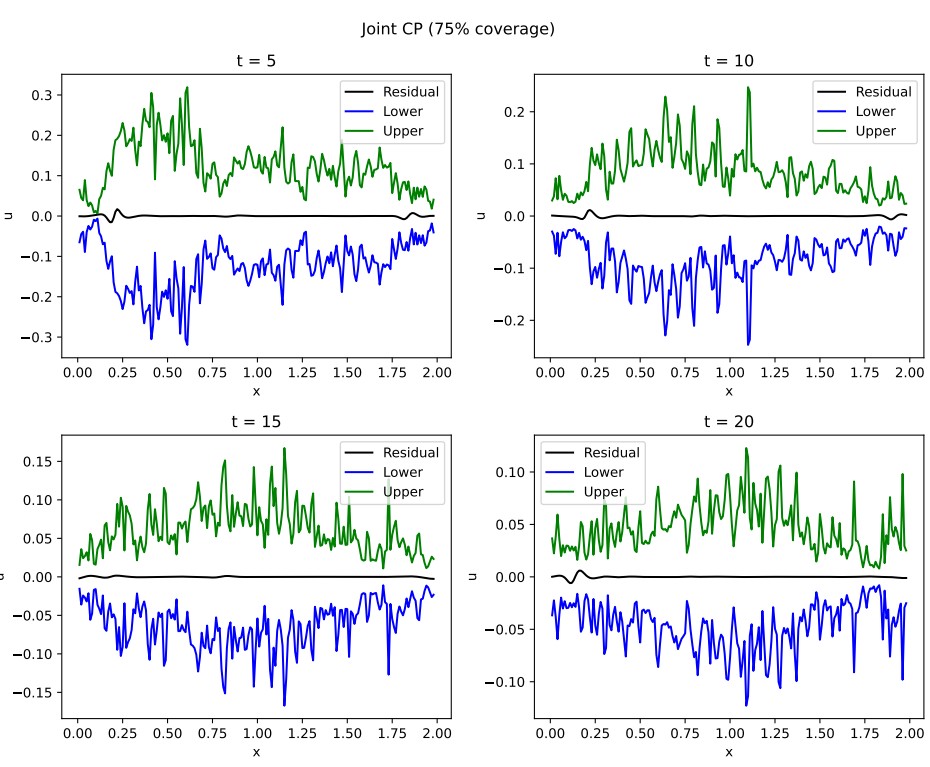

Figure 14: Burgers Equation: joint-CP with $\alpha = 0.75$

Table 8: Architecture of the 1D FNO deployed for modelling 1D Burgers' equation

| Part | Layer | Output Shape |
|------|-------|--------------|
| Input | - | (50, 1, 200, 1) |
| Lifting | Linear | (50, 1, 200, 32) |
| Fourier 1 | Fourier2d/Conv2d/Add/GELU | (50, 1, 32, 200) |
| Fourier 2 | Fourier2d/Conv2d/Add/GELU | (50, 1, 32, 200) |
| Fourier 3 | Fourier2d/Conv2d/Add/GELU | (50, 1, 32, 200) |
| Fourier 4 | Fourier2d/Conv2d/Add/GELU | (50, 1, 32, 200) |
| Fourier 5 | Fourier2d/Conv2d/Add/GELU | (50, 1, 32, 200) |
| Fourier 6 | Fourier2d/Conv2d/Add/GELU | (50, 1, 32, 200) |
| Projection 1 | Linear | (50, 1, 200, 256) |
| Projection 2 | Linear | (50, 1, 200 1) |

## G  WAVE EQUATION

### G.1  PHYSICS

Consider the two-dimensional wave equation:

$$\frac{\partial^2 u}{\partial t^2} = c^2\left(\frac{\partial^2 u}{\partial x^2} + \frac{\partial^2 u}{\partial y^2}\right) = 0, \quad x, y \in [-1, 1],\ t \in [0, 1],$$

$$u(x, y, t = 0) = e^{-A\left((x-X)^2 + (y-Y)^2\right)}, \tag{26}$$

$$\frac{\partial u(x, y, t = 0)}{\partial t} = 0, \qquad u(x, y, t) = 0, \quad x, y \in \partial\Omega,\ t \in [0, 1], \tag{27}$$

where $u$ defines the field variable, $c$ the wave velocity, $x$ and $y$ the spatial coordinates, $t$ the temporal coordinates. $A$, $X$ and $Y$ are variables that parameterise the initial condition of the PDE setup. There exists an additional constraint to the PDE setup that initialises the velocity of the wave to 0. The system is bounded periodically within the mentioned domain.

The solution for the wave equation is obtained by deploying a spectral solver that uses a leapfrog method for time discretisation and a Chebyshev spectral method on tensor product grid for spatial discretisation (Gopakumar et al., 2023a). The dataset is built by performing a Latin hypercube scan across the defined domain for the parameters $A, X, Y$, which accounts for the amplitude and the location of the 2D Gaussian, sampled for each simulation. We generate 1000 simulation points, each one with its initial condition and use it for training.

The physics of the equation, given by the various coefficients is held constant across the dataset generation throughout as given in eq. (26). Each data point, as in each simulation is generated with a different initial condition as described above. The parameters of the initial conditions are sampled from within the domain as given in table 9. Each simulation is run for 150-time iterations with a $\Delta t = 0.00667$ across a spatial domain spanning $[-1, 1]^2$, uniformly discretised into 64 spatial units in the x and y axes. The temporal domain is subsampled to factor in every $5^{th}$ time instance only.

Table 9: Domain range of initial condition parameters for the 2D wave equation.

| Parameter | Domain | Type |
|---|---|---|
| Amplitude $(A)$ | $[10, 50]$ | Continuous |
| X Position $(X)$ | $[0.1, 0.5]$ | Continuous |
| Y Position $(X)$ | $[0.1, 0.5]$ | Continuous |

### G.2  MODEL AND TRAINING

We train a 2D FNO to map the spatio-temporal evolution of the field variables. We deploy an auto-regressive structure that performs time rollouts allowing us to map the initial distribution recursively up until the $20^{th}$ time instance with a step size of 1. Each Fourier layer has 16 modes and a width of 32. The FNO architecture can be found in table 9. We employ a linear range normalisation scheme, placing the field values between -1 and 1. Each model is trained for up to 500 epochs using the Adam optimiser (Kingma & Ba, 2015) with a step-decaying learning rate. The learning rate is initially set to 0.005 and scheduled to decrease by half after every 100 epochs. The model was trained using an LP-loss (Gopakumar et al., 2024b). The performance of the trained model can be visualised in fig. 15.

### G.3  CALIBRATION AND VALIDATION

To perform the calibration as outlined in section 5, model predictions are obtained using initial conditions sampled from the domain given in table 9. The same bounded domain for the initial condition parameters is used for calibration and validation. 1000 initial conditions are sampled and fed to the model to perform the calibration and 100 samples are gathered for performing the validation.

Table 10: Architecture of the 2D FNO deployed for modelling the 2D wave equation

| Part | Layer | Output Shape |
|------|-------|--------------|
| Input | - | (50, 1, 64, 64, 1) |
| Lifting | `Linear` | (50, 1, 64, 64, 32) |
| Fourier 1 | `Fourier2d/Conv2d/Add/GELU` | (50, 1, 32, 64, 64) |
| Fourier 2 | `Fourier2d/Conv2d/Add/GELU` | (50, 1, 32, 64, 64) |
| Fourier 3 | `Fourier2d/Conv2d/Add/GELU` | (50, 1, 32, 64, 64) |
| Fourier 4 | `Fourier2d/Conv2d/Add/GELU` | (50, 1, 32, 64, 64) |
| Fourier 5 | `Fourier2d/Conv2d/Add/GELU` | (50, 1, 32, 64, 64) |
| Fourier 6 | `Fourier2d/Conv2d/Add/GELU` | (50, 1, 32, 64, 64) |
| Projection 1 | `Linear` | (50, 1, 64, 64, 256) |
| Projection 2 | `Linear` | (50, 1, 64, 64 1) |

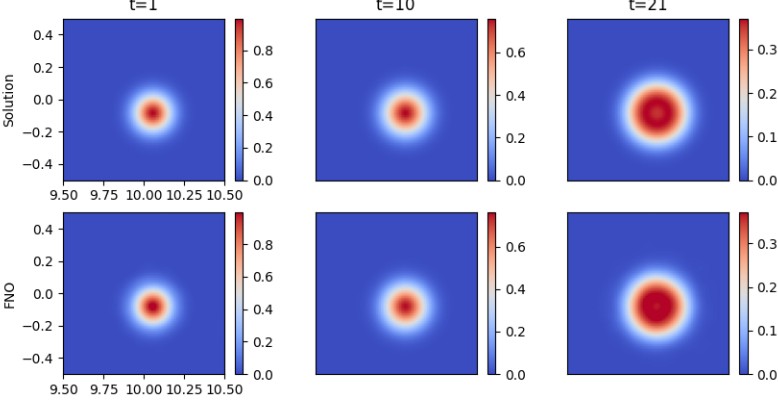

Figure 15: Wave Equation: Temporal evolution of field associated with the wave equation modelled using the numerical spectral solver (top of the figure) and that of the FNO (bottom of the figure). The spatial domain is given in Cartesian geometry.

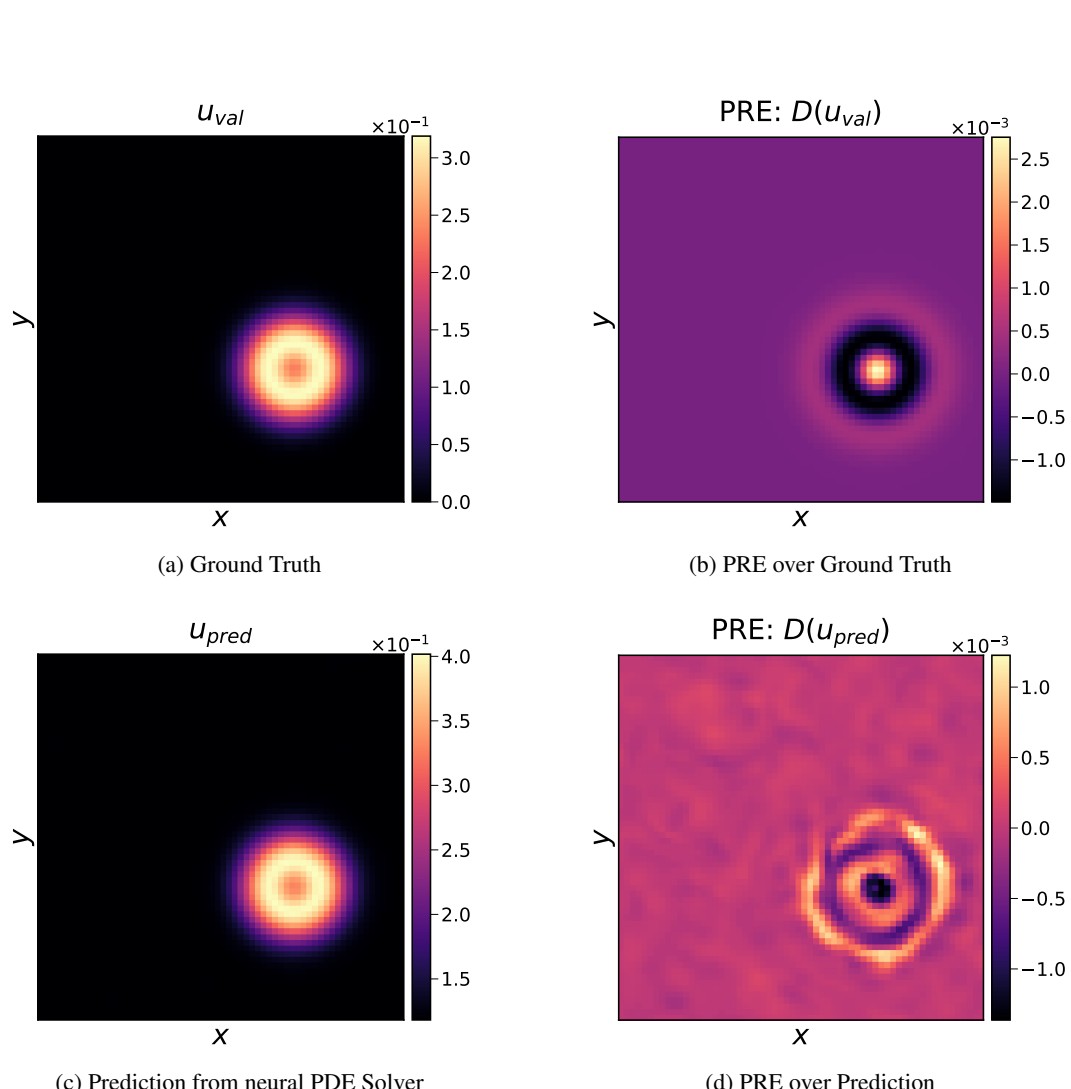

(a) Ground Truth

(b) PRE over Ground Truth

(c) Prediction from neural PDE Solver

(d) PRE over Prediction

Figure 16: Analying the PRE over the ground truth and the prediction. Though the neural PDE solver is capable of learning seemingly indistinguishable emulation of the physics while exploring the PRE over each tells a different story. As opposed to the smooth laplacian of the PRE over the ground truth, PRE over the prediction indicates a noisy solution, potentially arising due to the stochasticity of the optimisation process.

# H  2D NAVIER-STOKES EQUATIONS

## H.1  PHYSICS

Consider the two-dimensional Navier-Stokes equations:

$$\nabla \cdot \vec{\mathbf{v}} = 0,$$

$$\frac{\partial \vec{\mathbf{v}}}{\partial t} + (\vec{\mathbf{v}} \cdot \nabla)\vec{\mathbf{v}} = \nu \nabla^2 \vec{\mathbf{v}} - \nabla P,$$

with initial conditions:

$$u(x, y, t = 0) = -\sin(2\pi\alpha y) \quad y \in [-1, 1], \tag{28}$$
$$v(x, y, t = 0) = -\sin(4\pi\beta x) \quad x \in [-1, 1], \tag{29}$$

where $u$ defines the x-component of velocity, $v$ defines the y-component of velocity. The Navier-stokes equations solve the flow of an incompressible fluid with a kinematic viscosity $\nu$. The system is bounded with periodic boundary conditions within the domain. The dataset is built by performing a Latin hypercube scan across the defined domain for the parameters $\alpha, \beta$, which parameterises the initial velocity fields for each simulation. We generate 500 simulation points, each one with its initial condition and use it for training. The solver is built using a spectral method outlined in Philip Mocz's code.

Each data point, as in each simulation is generated with a different initial condition as described above. The parameters of the initial conditions are sampled from within the domain as given in table 11. Each simulation is run up until wallclock time reaches 0.5 $\Delta t = 0.001$. The spatial domain is uniformly discretised into 400 spatial units in the x and y axes. The temporal domain is subsampled to factor in every $10^{th}$ time instance, and the spatial domain is downsampled to factor every $4^{th}$ time instance leading to a $100 \times 100$ grid for the neural PDE.

Table 11: Domain range of initial condition parameters for the 2D Navier-Stokes equations

| Parameter | Domain | Type |
|---|---|---|
| Velocity x-axis ($u_0$) | $[0.5, 1.0]$ | Continuous |
| Velocity y-axis ($v_0$) | $[0.5, 1.0]$ | Continuous |

## H.2  MODEL AND TRAINING

We train a 2D multivariable FNO to map the spatio-temporal evolution of the field variables (Gopakumar et al., 2024b). We deploy an auto-regressive structure that performs time rollouts allowing us to map the initial distribution recursively up until the $20^{th}$ time instance with a step size of 1. Each Fourier layer has 8 modes and a width of 16. The FNO architecture can be found in table 12. We employ a linear range normalisation scheme, placing the field values between -1 and 1. Each model is trained for up to 500 epochs using the Adam optimiser (Kingma & Ba, 2015) with a step-decaying learning rate. The learning rate is initially set to 0.005 and scheduled to decrease by half after every 100 epochs. The model was trained using an LP-loss (Gopakumar et al., 2024b). The performance of the trained model can be visualised in fig. 17.

## H.3  CALIBRATION AND VALIDATION

To perform the calibration as outlined in section 5, model predictions are obtained using initial conditions sampled from the domain given in table 11. The same bounded domain for the initial condition parameters is used for calibration and validation. 1000 initial conditions are sampled and fed to the model to perform the calibration and 100 samples are gathered for performing the validation.

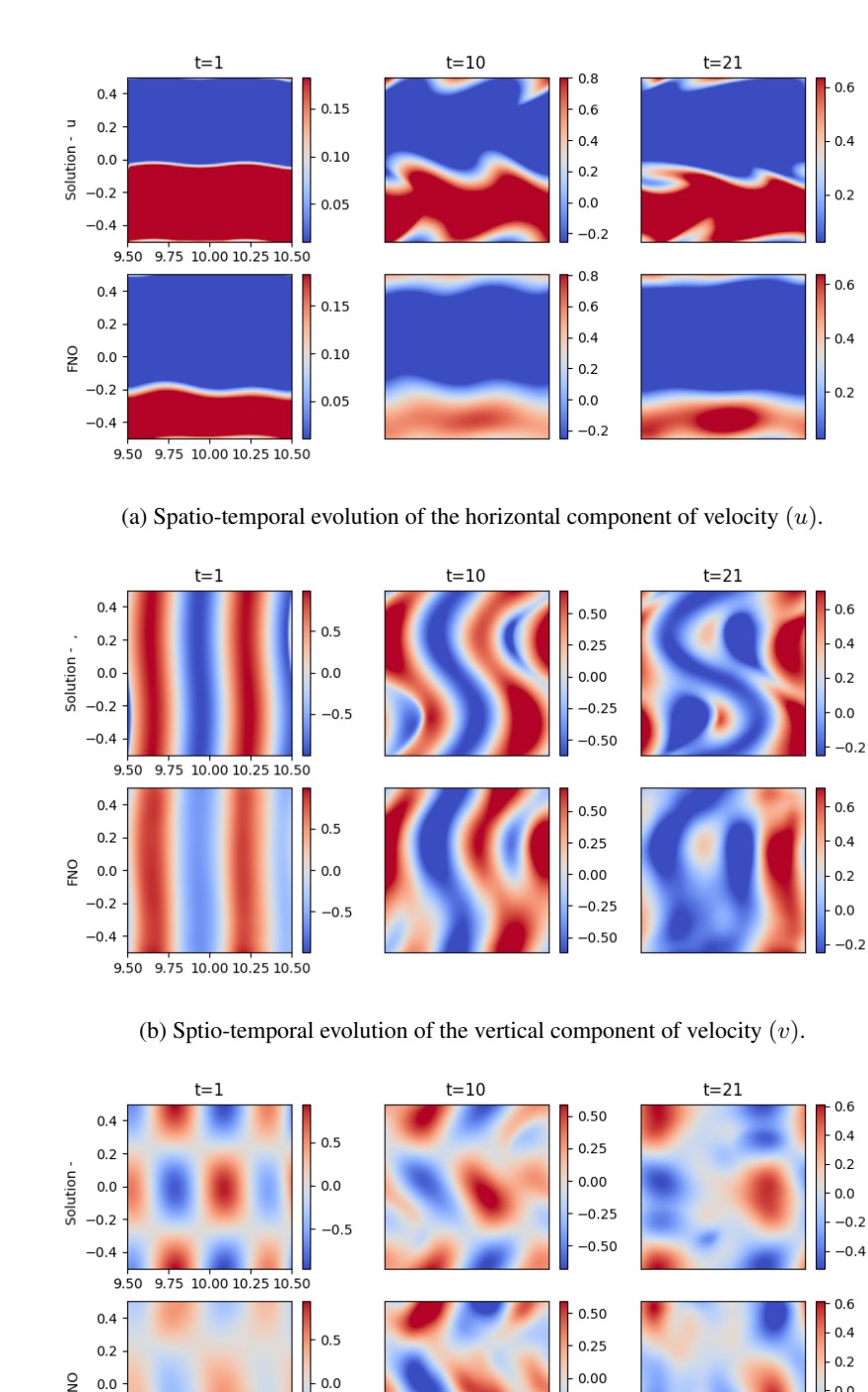

(a) Spatio-temporal evolution of the horizontal component of velocity $(u)$.

(b) Sptio-temporal evolution of the vertical component of velocity $(v)$.

(c) Spatio-temporal evolution of the pressure field $(P)$.

Figure 17: Navier-Stokes Equations: Temporal evolution of velocity and pressure modelled using the numerical spectral solver (top of the figure) and that of the FNO (bottom of the figure).

Table 12: Architecture of the 2D FNO deployed for modelling 2D Navier-Stokes equations

| Part | Layer | Output Shape |
|------|-------|--------------|
| Input | - | (50, 1, 100, 100, 1) |
| Lifting | `Linear` | (50, 1, 100, 100 16) |
| Fourier 1 | `Fourier2d/Conv2d/Add/GELU` | (50, 1, 16, 100, 100) |
| Fourier 2 | `Fourier2d/Conv2d/Add/GELU` | (50, 1, 16, 100, 100) |
| Fourier 3 | `Fourier2d/Conv2d/Add/GELU` | (50, 1, 16, 100, 100) |
| Fourier 4 | `Fourier2d/Conv2d/Add/GELU` | (50, 1, 16, 100, 100) |
| Fourier 5 | `Fourier2d/Conv2d/Add/GELU` | (50, 1, 16, 100, 100) |
| Fourier 6 | `Fourier2d/Conv2d/Add/GELU` | (50, 1, 16, 100, 100) |
| Projection 1 | `Linear` | (50, 1, 100, 100, 256) |
| Projection 2 | `Linear` | (50, 1, 100, 100 1) |

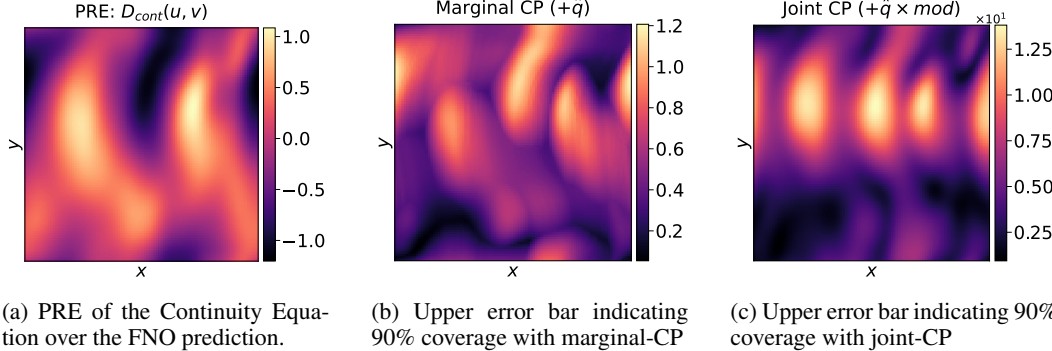

(a) PRE of the Continuity Equation over the FNO prediction.

(b) Upper error bar indicating 90% coverage with marginal-CP

(c) Upper error bar indicating 90% coverage with joint-CP

Figure 18: Navier-Stokes Equations: CP using the Continuity eq. (16) as the PRE.

# I  2D MAGNETOHYDRODYNAMICS

Consider the Ideal MHD equations in 2D:

$$\frac{\partial \rho}{\partial t} + \vec{\nabla} \cdot (\rho \vec{v}) = 0,$$

$$\rho\left(\frac{\partial \vec{v}}{\partial t} + \vec{v} \cdot \nabla \vec{v}\right) = \frac{1}{\mu_0}\vec{B} \times (\vec{\nabla} \times \vec{B}) - \nabla P,$$

$$\frac{d}{dt}\left(\frac{P}{\rho^\gamma}\right) = 0,$$

$$\frac{\partial \vec{B}}{\partial t} = \vec{\nabla} \times (\vec{v} \times \vec{B}),$$

$$\vec{\nabla} \cdot \vec{B} = 0,$$

with initial conditions:

$$u = -sin(2a\pi Y), \tag{30}$$

$$v = sin(2b\pi X), \tag{31}$$

$$P = \frac{\gamma}{4c\pi}, \tag{32}$$

where the density ($\rho$), velocity field ($\vec{v} = [u, v]$) and the pressure of plasma is modelled under a magnetic field ($\vec{B} = [B_x, B_y]$) across a spatio-temporal domain $x, y \in [0, 1]^2$, $t \in [0, 5]$. $\mu_0$ is taken to be the magnetic permeability of free space. The system is bounded with periodic boundary conditions within the domain. The dataset is built by performing a Latin hypercube scan across the defined domain for the parameters $a, b, c$, which parameterises the initial velocity fields for each simulation. We generate 500 simulation points, each one with its initial condition and use it for training. The solver is built using a finite volume method outlined in Philip Mocz's code.

Each data point, as in each simulation is generated with a different initial condition as described above. The parameters of the initial conditions are sampled from within the domain as given in table 11. Each simulation is run up until wallclock time reaches 0.5 with a varying temporal discretisation. The spatial domain is uniformly discretised into 128 spatial units in the x and y axes. The temporal domain is downsampled to factor in every $25^{th}$ time instance.

Table 13: Domain range of initial condition parameters for the 2D MHD equations

| Parameter | Domain | Type |
|---|---|---|
| Velocity x-axis ($a$) | $[0.5, 1.0]$ | Continuous |
| Velocity y-axis ($b$) | $[0.5, 1.0]$ | Continuous |
| Pressure ($c$) | $[0.5, 1.0]$ | Continuous |

## I.1  MODEL AND TRAINING

We train a 2D multi-variable FNO to map the spatio-temporal evolution of the 6 field variables collectively. We deploy an auto-regressive structure that performs time rollouts allowing us to map the initial distribution recursively up until the $20^{th}$ time instance with a step size of 1. Each Fourier layer has 8 modes and a width of 16. The FNO architecture can be found in table 14. We employ a linear range normalisation scheme, placing the field values between -1 and 1. Each model is trained for up to 500 epochs using the Adam optimiser (Kingma & Ba, 2015) with a step-decaying learning rate. The learning rate is initially set to 0.005 and scheduled to decrease by half after every 100 epochs. The model was trained using an LP-loss (Gopakumar et al., 2024b). The performance of the trained model can be visualised in fig. 19.

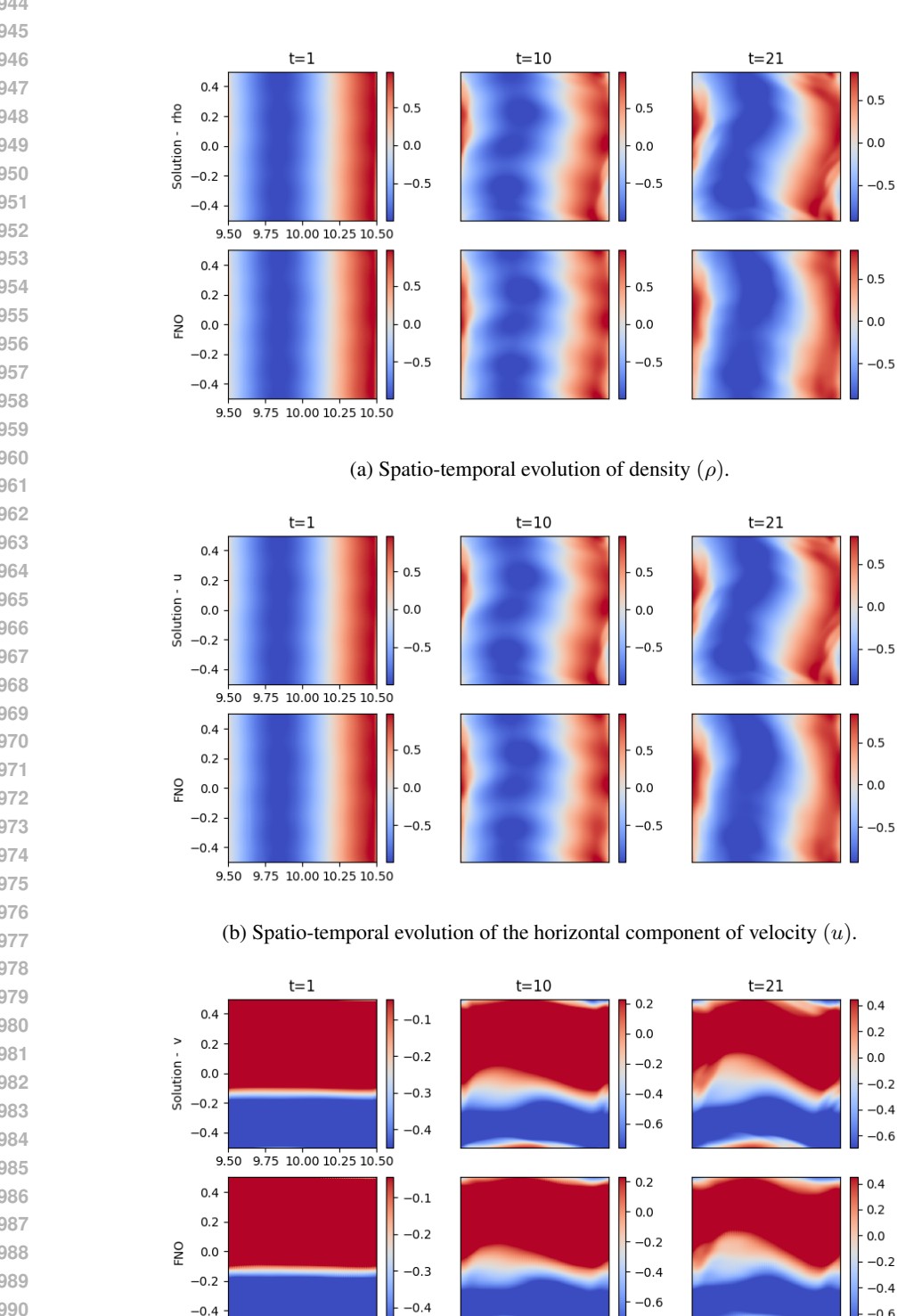

(a) Spatio-temporal evolution of density ($\rho$).

(b) Spatio-temporal evolution of the horizontal component of velocity ($u$).

(c) Spatio-temporal evolution of the vertical component of velocity ($v$).

Figure 19: Navier-Stokes Equations: Temporal evolution of velocity and pressure modelled using the numerical spectral solver (top of the figure) and that of the FNO (bottom of the figure). (Continued on next page)

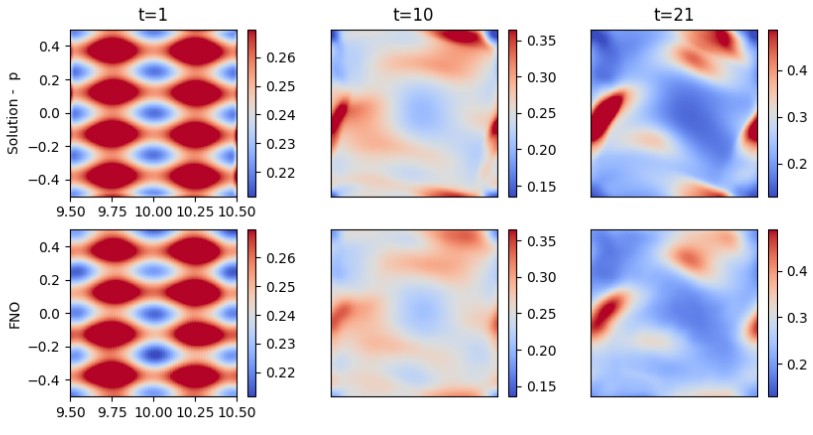

(d) Spatio-temporal evolution of the pressure field $(P)$.

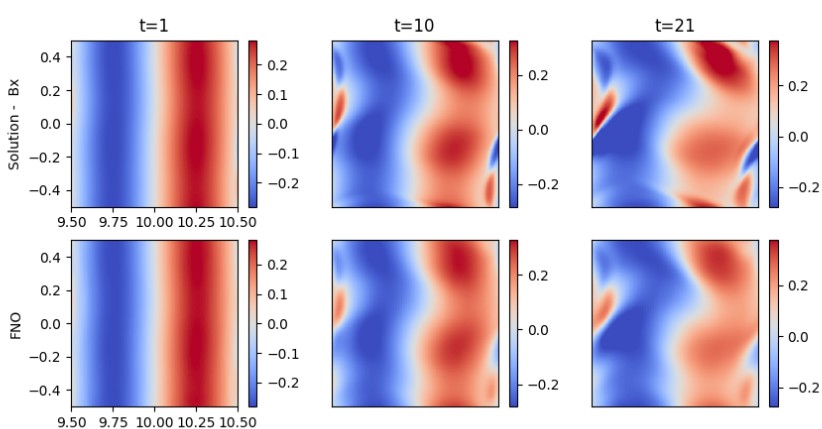

(e) Spatio-temporal evolution of the horizontal component of the magnetic field $(B_x)$.

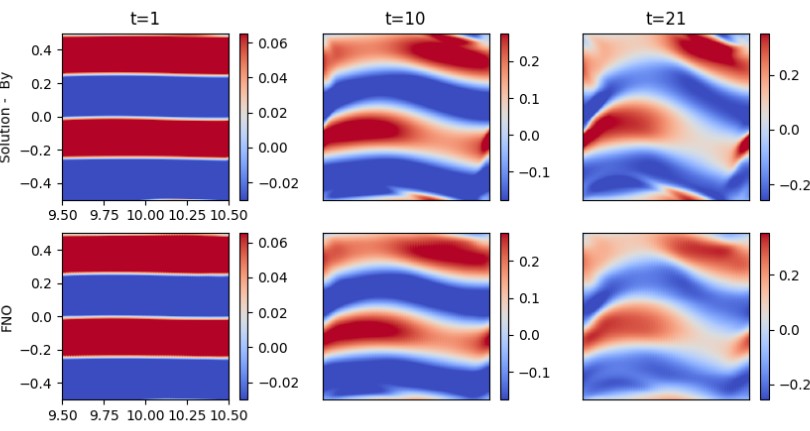

(f) Spatio-temporal evolution of the vertical component of the magnetic field $(B_y)$.

Figure 19: Navier-Stokes Equations: Temporal evolution of velocity and pressure modelled using the numerical spectral solver (top of the figure) and that of the FNO (bottom of the figure). (Continued from previous page)

Table 14: Architecture of the 2D FNO deployed for modelling 2D MHD equations

| Part | Layer | Output Shape |
|------|-------|--------------|
| Input | - | (50, 1, 128, 128, 1) |
| Lifting | Linear | (50, 1, 128, 128 16) |
| Fourier 1 | Fourier2d/Conv2d/Add/GELU | (50, 1, 16, 128, 128) |
| Fourier 2 | Fourier2d/Conv2d/Add/GELU | (50, 1, 16, 128, 128) |
| Fourier 3 | Fourier2d/Conv2d/Add/GELU | (50, 1, 16, 128, 128) |
| Fourier 4 | Fourier2d/Conv2d/Add/GELU | (50, 1, 16, 128, 128) |
| Fourier 5 | Fourier2d/Conv2d/Add/GELU | (50, 1, 16, 128, 128) |
| Fourier 6 | Fourier2d/Conv2d/Add/GELU | (50, 1, 16, 128, 128) |
| Projection 1 | Linear | (50, 1, 128, 128, 256) |
| Projection 2 | Linear | (50, 1, 128, 128 1) |

## I.2 CALIBRATION AND VALIDATION

To perform the calibration as outlined in section 5, model predictions are obtained using initial conditions sampled from the domain given in table 11. The same bounded domain for the initial condition parameters is used for calibration and validation. 100 initial conditions are sampled and fed to the model to perform the calibration and 100 samples are gathered for validation.

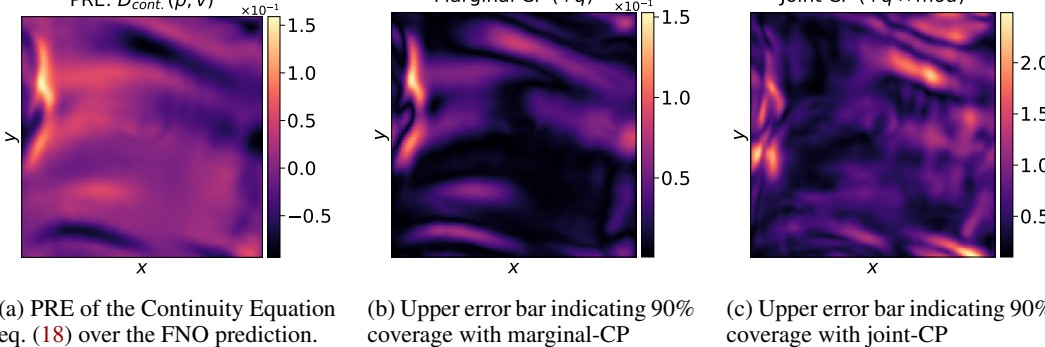

(a) PRE of the Continuity Equation eq. (18) over the FNO prediction.

(b) Upper error bar indicating 90% coverage with marginal-CP

(c) Upper error bar indicating 90% coverage with joint-CP

Figure 20: MHD: CP using the Continuity Equation (18) as the PRE for a neural PDE surrogate model solving the Ideal MHD equations.

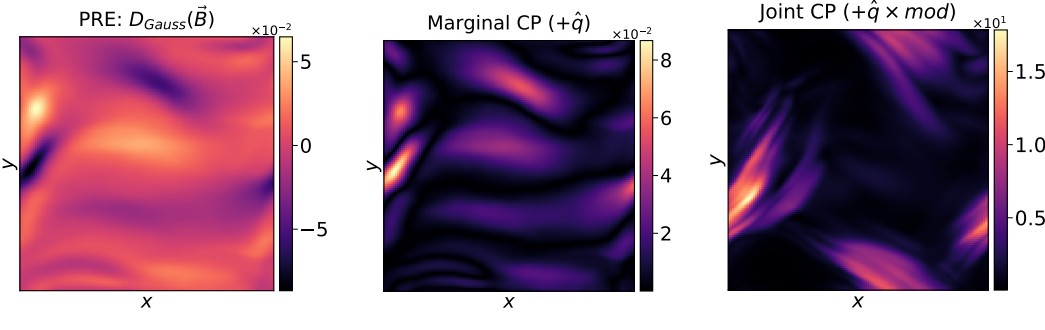

(a) PRE of the Induction Equation eq. (22) over the FNO prediction.

(b) Upper error bar indicating 90% coverage with marginal-CP

(c) Upper error bar indicating 90% coverage with joint-CP

Figure 21: MHD: CP using the Gauss's law for magnetism eq. (22) as the PRE for a neural PDE surrogate model solving the Ideal MHD equations.

## J  INITIAL AND BOUNDARY CONDITIONS

As mentioned in section 4.3, the focus of our experiments has been in quantifying the misalignment of the model with the PDE in the domain of the problem. A well-defined PDE is characterised by the PDE on the domain, the initial condition across the domain at $t = 0$ and the boundary conditions, reflecting the physics at the boundary. Within a neural-PDE setting, the initial condition does not need to be enforced or measured for as the neural-PDE is set up as an initial-value problem, taking in the initial state to autoregressively evolve the later timesteps and hence does not come under the purview of the neural-PDE's outputs. The boundary conditions, whether Dirichlet, Neumann or periodic, follows a residual structure as outlined in eq. (2), allowing us to use it as a PRE-like nonconformity score for performing conformal prediction. In all the problems we have under consideration, the PDEs are modelled under periodic boundary conditions:

$$\frac{\partial u}{\partial X} = 0; \ X \in \partial \Omega \tag{33}$$

By deploying the eqn 33 as the PRE across the boundary, we can obtain error bars over the boundary conditions as well. Within fig. 22, we demonstrate the error bars obtained by using the boundary conditions as the PRE nonconformity scores for the Navier-Stokes equations.

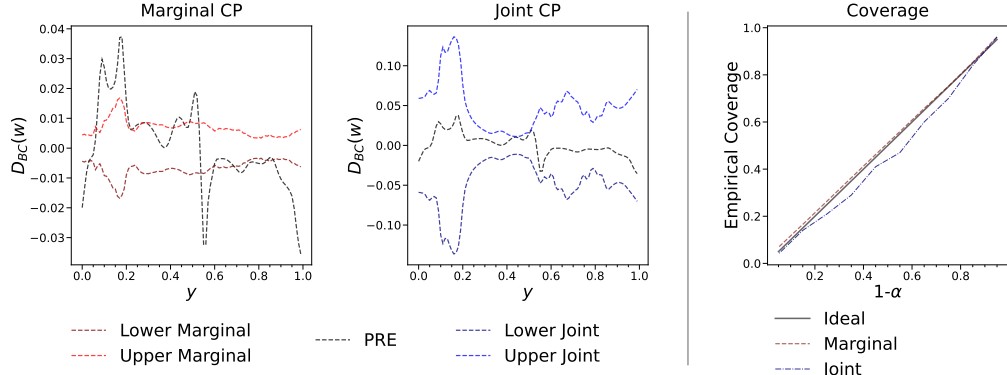

Figure 22: Error bars obtained over the boundary conditions over the right wall of domain of the Navier-Stokes Equation using Marginal and Joint CP. The empirical coverage obtained using the boundary condition as the PRE nonconformity score is also given.

## K    UTILISING PRE-CP AS A MEASURE OF MODEL QUALITY

While evaluating the performance of a neural-PDE, it is important to their fit not just to the data but to the underlying physics. PRE-CP will provide guaranteed coverage irregardless of the quality of the model. It will have considerably wider error bounds when the neural-PDE (whether PINN or a Neural Operator) fails to comply with the physics. However, we believe that this is an advantage of our method. In PRE-CP formulation, the bounds are estimated across the PDE residual, where the ground truth for a well-fit solution should always be near zero. If we get wide error bars further away from the 0 for potentially high coverage estimates, it is a strong indication that statistically the model violates the physics of interest.

Consider the example with the Advection equation. We have two models, a well-fit (good model) and a poorly fit one (bad model). As shown in fig. 23, though we obtain guaranteed coverage in the case of both the bad and good models, the width of the error bars indicates the quality of the model. Taken for 90 % coverage, the width of the coverage bounds obtained over the bad model is substantially larger than that obtained by the good model.

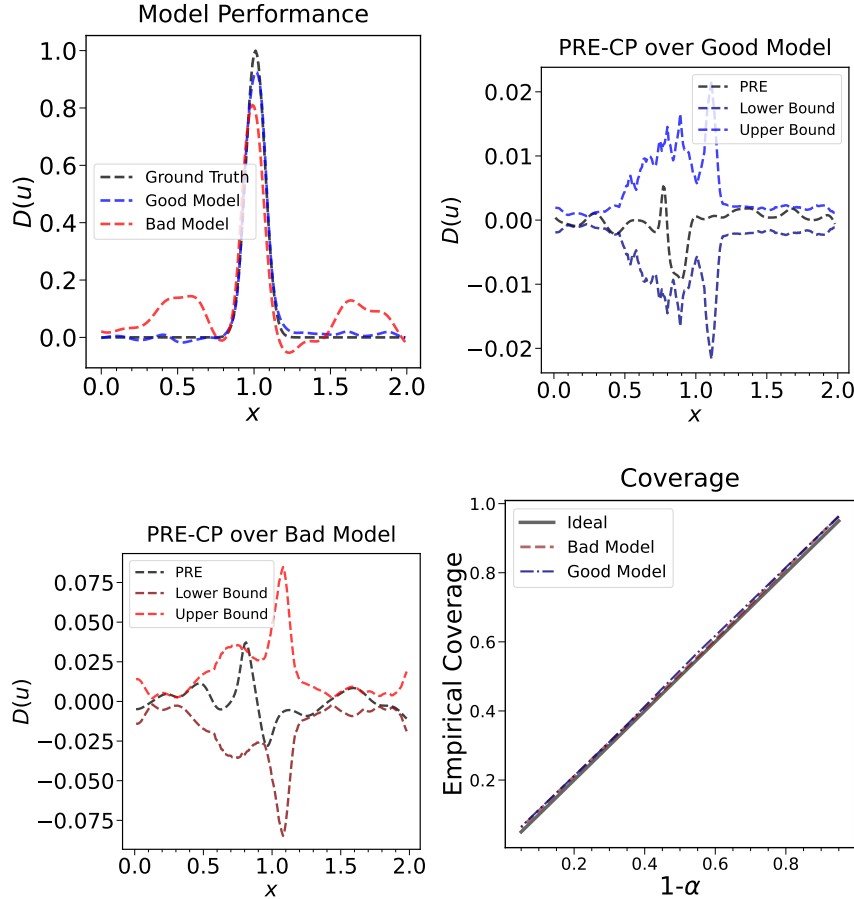

Figure 23: PRE-CP provides guaranteed coverage irrespective of the model performance, however, the width of the obtained coverage bounds indicates the accuracy of the model in obeying the underlying physics. Coverage taken for $\alpha = 0.1 \sim 90\%$ coverage.

There still could be a concern as to what width can be considered to be within a good range within the residual space. This could be estimated by running the PRE convolution operator(s) across a single numerical simulation of the interested physics, thereby estimating the impact of the operator in estimating the residual. The PRE over the simulation data will allow us to judge what ranges for the coverage width differentiate between a "bad" and a "good" model.

## L    PLASMA MODELLING WITHIN A TOKAMAK

In (Gopakumar et al., 2024b), the authors model the evolution of plasma blobs within a Fusion reactor (known as a Tokamak) following reduced magnetohydrodynamics. Plasma, characterised by density $\rho$, electric potential $\phi$ and Temperature $T$ under the absence of magnetic pressure confining it in place, moves radially outward to the wall of the reactor driven by its kinetic pressure. We demonstrate the ability to scale our method by applying it to obtain valid error bars across the multi-variable FNO trained for plasma modelling.

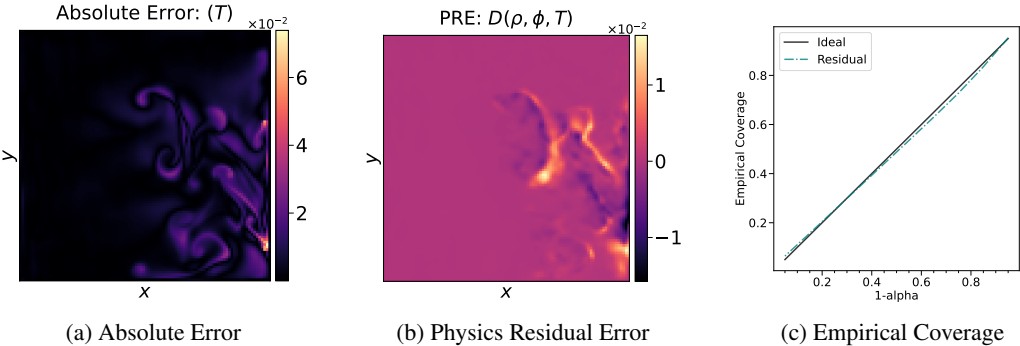

(a) Absolute Error       (b) Physics Residual Error       (c) Empirical Coverage

Figure 24: **Reduced MHD:** PRE-CP using the Temperature equation (Eqn. 3 in (Gopakumar et al., 2024b)) of reduced-MHD to bound the plasma surrogate models. The PRE captures the model error relatively well, allowing us to provide lower and upper error bars corresponding to our required coverage.

As shown in figure 24, our method can capture the model error across a range of predictions and can devise error bars that provide guaranteed coverage without needing any additional data. In figure 24a, we demonstrate the absolute error in the model prediction of the temperature evolution, and correlate that with the PRE over the temperature equations in figure 24b.

