# OpenReview forum: "Calibrated Physics-Informed Uncertainty Quantification"
_ICLR.cc/2025/Conference — Submitted to ICLR 2025_

### Official Review · Reviewer_kq4A · 2024-10-29

**Soundness:** 3
**Presentation:** 2
**Contribution:** 2
**Rating:** 6
**Confidence:** 2

**Summary:**

The paper introduces a method to quantify the prediction uncertainty of a data-driven neural PDE solver using conformal prediction. This technique leverages the assumption that the physical residual operator, which defines the PDE dynamics, is known. By using a discretized version of this operator, the method assesses whether the neural solver’s predictions are consistent with the expected dynamics.

**Strengths:**

- The paper introduces a novel approach that integrates a physical prior with uncertainty quantification techniques to evaluate the performance of purely data-driven neural PDE solvers.
- The idea is presented with clear details and supported by experiments on diverse equations. The authors effectively illustrate how this helps assess and improve the reliability of the neural PDE predictions.

**Weaknesses:**

- My primary concern is about the choice to use only the physical residual for PRE non-conformity. Since most neural PDE solvers for temporal dynamics aim to solve an Initial-Boundary Value Problem (IBVP), it's essential not only to consider the differential operator but also to account for how the solution aligns with the imposed initial and boundary conditions. The authors may need to address this aspect of the problem and explore how incorporating these constraints could enhance the conformity assessment.

**Questions:**

I suggest that the authors provide a more intuitive explanation of how the conformal prediction techniques applied in the paper work. Offering clearer insights into these methods could help a broader audience better understand the concepts and their relevance to the study.

---

> ### Author Response · Authors · 2024-11-22
> **Initial and Boundary Conditions**
>
> We thank the reviewer for this interesting question and for giving us the opportunity to elaborate further on an interesting point.
>
> Though our focus within this paper has been on quantifying the misalignment of the model with the equations in the domain of the problem (as mentioned in section 4.3), we should account for the initial and boundary conditions.
>
> A well-defined PDE is characterised by the equations on the domain, the initial condition across the domain at $t=0$ and the boundary conditions, reflecting the physics at the boundary. Within a neural-PDE setting, the initial condition does not need to be enforced or measured for as the neural-PDE is set up as an initial-value problem, taking in the initial state to autoregressively evolve the later time-steps and hence does not come under the purview of the neural-PDE's outputs. The boundary conditions, whether Dirichlet, Neumann or periodic, follow a residual structure as outlined in eqn 2 within the paper, allowing us to use it as a PRE-like nonconformity score for performing conformal prediction. In all the problems we have under consideration, the PDEs are modelled under periodic boundary conditions:
>
> $$\frac{\partial u }{\partial X} = 0 ;   X \in \partial \Omega $$
>
> By deploying the above equation as the PRE across the boundary, we can obtain error bars over the boundary conditions as well. Within figure 21 of the revised paper, we demonstrate the error bars obtained by using the boundary conditions as the PRE nonconformity scores for the Navier-Stokes equations.
>
> We have included the treatment of boundary conditions as a section within the appendix (Appendix J).

---

> ### Author Response · Authors · 2024-11-22
> **Intuitive Explanation**
>
> That's a brilliant suggestion. We have crafted a more intuitive explanation here.
>
> Our framework makes uncertainty quantification for neural-PDE solvers more intuitive and practically useful by leveraging two key insights. First, we use the PDE residual (how well the solution satisfies the underlying physics equations) as a measure of uncertainty, rather than traditional error metrics that require ground truth data. This means we can evaluate the quality of predictions without needing expensive simulation data - if a predicted solution violates conservation laws by a large amount, we know it's likely to be inaccurate. Second, we calibrate these physics-based error estimates using conformal prediction, which provides statistical guarantees about the uncertainty bounds. This is like having a physics-aware "confidence score" that tells us how much we can trust each prediction.
>
> The practical value becomes clear in applications - imagine using a neural-PDE solver for weather forecasting or fusion reactor control. Our method provides calibrated error bounds that tell you when predictions are physically implausible and should be double-checked with more expensive traditional methods. We offer two complementary ways to quantify uncertainty: "marginal" coverage that gives error bars for each point in space and time (useful for identifying specific problematic regions), and "joint" coverage that evaluates entire predictions (useful for filtering out globally unreliable solutions). This makes the uncertainty estimates both theoretically rigorous and practically actionable for scientists and engineers using these models.

---

> ### Author Response · Authors · 2024-11-27
>
> We thank the reviewer for the helpful comments and the opportunity for further discussion. As we are approaching the last day allowed for PDF revision, could you kindly let us know if there are any other concerns that we should address?

---

### Official Review · Reviewer_YEc3 · 2024-10-31

**Soundness:** 1
**Presentation:** 2
**Contribution:** 1
**Rating:** 3
**Confidence:** 4

**Summary:**

The paper uses physics residual errors as nonconformity scores within a conformal prediction framework for efficient and calibrated uncertainty quantification in neural PDEs.

**Strengths:**

The paper is clearly written and easy to understand.

**Weaknesses:**

1. The proposed method appears relatively simple and lacks novelty. Using the Physics Residual Error as a nonconformity score for conformal prediction seems to be a minor contribution.
2. The paper lacks a comparison with other uncertainty quantification (UQ) methods. I think various UQ methods could be applied based on the point-wise PDE residuals.
3. Additional analytical experiments are necessary to gain a deeper understanding of the proposed method. Could you conduct some tests in real-world scenarios?
4. The numerical PDE residual shows considerable numerical error due to discretization issues, particularly in 2D+time PDEs. How is the time derivative calculated over a 20-frame sequence? Please provide further clarification.

**Questions:**

See Weaknesses

---

> ### Author Response · Authors · 2024-11-22
> **Lack of Novelty**
>
> We would like to humbly disagree with this statement as we believe that there is a novelty in the nonconformity score and how that extends into performing data-free CP. As reviewer mcYW points out, the paper touches on important topics within the SciML community and we believe that this work leads to practical applications of UQ and active learning. The PRE allows to obtaining calibrated uncertainties for neural operator models, while current UQ methods for such models do not provide guaranteed coverage. The calibration is data-free, which addresses the well-known usual limitation of conformal prediction, which for other nonconformity scores relies on potentially expensive calibration data. In the table comparing various features of different UQ methods (Table 1 in Appendix C of the revised paper), the novelty of our work is further highlighted.

---

> ### Author Response · Authors · 2024-11-22
> **Comparison to other UQ methods**
>
> We did not include comparisons to other UQ methods as our method involved quantifying the physics residual error and obtaining its bounds as opposed to directly quantifying the model's predictive error. However, considering the concerns raised by the reviewer regarding providing baseline comparisons and the effectiveness such a comparison provides to our claims, we have conducted experiments to demonstrate how our method stands out as being data-free, not requiring any modifications or sampling, and helps obtain guaranteed coverage bounds in a physics-informed manner. We have added them to the revised paper under Appendix C.

---

> > ### Author Response · Authors · 2024-11-22
> > **Qualitiative Comparison**
> >
> > | Method        | Data-Free | Modification-Free | Sampling-Free | Guaranteed Coverage | Physics-Informed |
> > |---------------|-----------|-------------------|---------------|---------------------|------------------|
> > | MC Dropout    | ✓         | ✗                 | ✗             | ✗                   | ✗                |
> > | Deep Ensemble | ✓         | ✗                 | ✗             | ✗                   | ✗                |
> > | BNN           | ✓         | ✗                 | ✗             | ✗                   | ✗                |
> > | SWA-G         | ✓         | ✗                 | ✗             | ✗                   | ✗                |
> > | CP-AER        | ✗         | ✓                 | ✓             | ✓                   | ✗                |
> > | CP-PRE (Ours) | ✓         | ✓                 | ✓             | ✓                   | ✓                |
> >
> >  Our method is data-free, does not require any modifications or sampling, and helps obtain guaranteed coverage bounds in a physics-informed manner.

---

> ### Author Response · Authors · 2024-11-22
> **Experiment in Real-world Scenario**
>
> Within this experiment, we explore the modelling of plasma for Nuclear Fusion, where surrogate models are being used to determine the real-time stability of the reactor. These surrogate models mimic complex numerical codes (such as JOREK / STORM) and predict whether the reactor would enter into a catastrophic disruption event. Hence, being able to provide inexpensive and immediate yet valid error bars is of utmost urgency as it allows you to isolate events and prevent significant damage to the reactor.
>
> In [1], the authors model the evolution of plasma blobs within a Fusion reactor (known as a Tokamak) following reduced magnetohydrodynamics. Plasma, characterised by density $\rho$, electric potential $\phi$ and Temperature $T$ under the absence of magnetic pressure confining it in place, moves radially outward to the wall of the reactor driven by its kinetic pressure.  We demonstrate the ability to scale our method by applying it to obtain valid error bars across the multi-variable FNO trained for plasma modelling.
>
> As shown in figure  23 of the revised paper, our method can capture the model error across a range of predictions and can devise error bars that provide guaranteed coverage without needing any additional data. In figure 23(a), we demonstrate the absolute error in the model prediction of the temperature evolution, and correlate that with the PRE over the temperature equations in figure 23(b). The details of the new experiment has been added to Appendix L of the revised paper.
>
> [1] V. Gopakumar, S. Pamela, L. Zanisi, Z. Li, A. Gray, D. Brennand, N. Bhatia, G. Stathopoulos, M. Kusner, M. P. Deisenroth, A. Anandkumar, the JOREK Team, and MAST Team, "Plasma surrogate modelling using Fourier neural operators," Nuclear Fusion, vol. 64, no. 5, p. 056025, Apr. 2024, doi: 10.1088/1741-4326/ad313a

---

> ### Author Response · Authors · 2024-11-22
> **Numerical Errors due to Discretisation**
>
> We would like to thank the reviewer for this question as it puts emphasis on an important aspect of neural-PDEs and our method.
>
> As demonstrated in [1], the discretisation of the inputs and hence model outputs plays an important role in the accuracy of the neural-PDE solvers. Though the neural operators are constructed for discretisation-invariant behaviour due to the band-limited nature of the functions, they often exhibit discretisation-convergent behaviour rather than be fully discretisation-invariant. This is of particular importance in the temporal dimensions as these neural-PDE models utilise a discrete, autoregressive-based time-stepping and is baked into the model within its training regime [2]. Due to lack of control in the discretisation within the temporal domain $(dt)$, the PRE estimates tend to have higher numerical errors as well. In figure 9 of the revised paper,  we visualise the evaluation of finite difference in 2D+time as a 3D convolution. The finite difference stencil i.e. the convolutional kernel has a unit discretisation of $dx$, $dy$ and $dt$ associated with the problem and is applied over the signal i.e. the output from the neural-PDE $u$ spanning the domain $x,y,t$, where $x \in [0, X], \\; y \in [0, Y], \\; t \in [0, T] $.
>
> We thank the reviewer for raising this discussion and believe that this represents an important component of using our method and have added a subsection explaining the role of discretisation in the appendix D.
>
> [1] F. Bartolucci, E. de Bezenac, B. Raonic, R. Molinaro, S. Mishra, and R. Alaifari, ``Representation Equivalent Neural Operators: a Framework for Alias-free Operator Learning,'' in Proceedings of the 37th Conference on Neural Information Processing Systems (NeurIPS), 2023
>
> [2] M. McCabe, P. Harrington, S. Subramanian, and J. Brown, ``Towards Stability of Autoregressive Neural Operators,'' Transactions on Machine Learning Research, 2023

---

> ### Author Response · Authors · 2024-11-22
> **Quantitative Comparison**
>
> ## `Wave Equation - Coverage measured for 2σ(∼ 95%)`
>
> #
>
> | UQ            | L2 (in-dist.)           | Coverage (in-dist.) | L2 (out-dist.)          | Coverage (out-dist.) | Train time (hr) | Eval. time (s) |
> |:-------------:|:-----------------------:|:-------------------:|:-----------------------:|:--------------------:|:---------------:|:--------------:|
> | Deterministic | 1.77e-05 $\pm$ 3.69e-07 | -                   | 2.46e-03 $\pm$ 2.00e-05 | -                    | 0:38            | 22             |
> | MC Dropout    | 1.44e-04 $\pm$ 3.26e-06 | 97.31 $\pm$ 0.03    | 2.12e-03 $\pm$ 2.60e-05 | 89.83 $\pm$ 0.07     | 0:52            | 120            |
> | Deep Ensemble | 8.76e-06 $\pm$ 2.43e-07 | 98.02 $\pm$ 0.04    | 2.42e-03 $\pm$ 1.58e-05 | 83.44 $\pm$ 0.12     | 3:10            | 112            |
> | BNN           | 1.92e-04 $\pm$ 1.92e-06 | 97.10 $\pm$ 0.09    | 2.67e-03 $\pm$ 1.26e-05 | 91.76 $\pm$ 0.10     | 0:53            | 118            |
> | SWA-G         | 1.41e-05 $\pm$ 1.74e-06 | 94.55 $\pm$ 3.25    | 2.55e-03 $\pm$ 2.82e-05 | 81.90 $\pm$ 3.31     | 0:47            | 113            |
> | CP-AER        | 1.76e-05 $\pm$ 4.40e-07 | 95.70 $\pm$ 0.21    | 2.46e-03 $\pm$ 1.41e-05 | 95.59 $\pm$ 0.14     | 0:38            | 23             |
> | CP-PRE (Ours) | 1.78e-05 $\pm$ 4.61e-07 | 95.52 $\pm$ 0.21    | 2.46e-03 $\pm$ 1.25e-05 | 95.39 $\pm$ 0.12     | 0:38            | 23             |
>
> #
> ## `Navier-Stokes Equations - Coverage measured for 2σ(∼ 95%)`
>
> #
>
>
> | UQ            | L2 (in-dist.)           | Coverage (in-dist.) | L2 (out-dist.)           | Coverage (out-dist.) | Train time (hr) | Eval. time (s) |
> |:-------------:|:-----------------------:|:-------------------:|:------------------------:|:--------------------:|:---------------:|:--------------:|
> | Deterministic | 1.05e-04 $\pm$ 6.91e-06 | -                   | 3.67e-03 $\pm$ 5.30e-05  | -                    | 3:22            | 25             |
> | MC Dropout    | 5.96e-04 $\pm$ 2.30e-05 | 82.21 $\pm$ 0.22    | 4.30e-03 $\pm$ 8.05e-05  | 44.05 $\pm$ 0.26     | 3:34            | 153            |
> | Deep Ensemble | 1.22e-04 $\pm$ 3.95e-06 | 91.31 $\pm$ 0.08    | 3.67e-03 $\pm$ 3.52e-05  | 30.74 $\pm$ 0.19     | 16:22           | 147            |
> | BNN           | 6.90e-03 $\pm$ 1.31e-04 | 89.91 $\pm$ 0.20    | 6.95e-03 $\pm$  1.31e-04 | 85.19 $\pm$ 0.23     | 3:39            | 152            |
> | SWA-G         | 1.96e-04 $\pm$ 1.15e-05 | 84.22 $\pm$ 2.37    | 3.63e-03 $\pm$ 1.37e-04  | 31.00 $\pm$ 2.85     | 3:28            | 146            |
> | CP-AER        | 1.05e-04 $\pm$ 6.58e-06 | 95.56 $\pm$ 0.40    | 3.66e-03 $\pm$ 2.81e-05  | 95.54 $\pm$ 0.15     | 3:22            | 26             |
> | CP-PRE (Ours) | 1.07e-04 $\pm$ 5.18e-06 | 95.44 $\pm$ 0.22    | 3.70e-03 $\pm$ 4.23e-05  | 95.57 $\pm$ 0.14     | 3:22            | 34             |
>
>
> #
> ## `MHD Equations - Coverage measured for 2σ(∼ 95%)`
>
> #
> | UQ            | L2 (in-dist.)            | Coverage (in-dist.) | L2 (out-dist.)          | Coverage (out-dist.) | Train time (hr) | Eval. time (s) |
> |:-------------:|:------------------------:|:-------------------:|:-----------------------:|:--------------------:|:---------------:|:--------------:|
> | Deterministic | 2.20e-03 $\pm$ 5.20e-03  | -                   | 4.71e-02 $\pm$ 1.06e-03 | -                    | 5:00            | 40             |
> | MC Dropout    | 3.29e-02 $\pm$ 5.86e-04  | 41.13 $\pm$ 0.19    | 2.09e-01 $\pm$ 1.38e-03 | 16.91 $\pm$ 0.06     | 5:30            | 240            |
> | Deep Ensemble | 3.59e-03 $\pm$ 3.51e-04  | 78.15 $\pm$ 0.16    | 3.41e-01 $\pm$ 3.15e-02 | 39.63 $\pm$ 0.31     | 26:25           | 235            |
> | BNN           | 4.20e-03 $\pm$  4.08e-05 | 90.24 $\pm$ 0.10    | 4.63e-02 $\pm$ 8.98e-04 | 62.37 $\pm$ 0.46     | 5:40            | 240            |
> | SWA-G         | 2.61e-03 $\pm$ 9.68e-05  | 48.50 $\pm$ 3.81    | 4.53e-02 $\pm$ 6.64e-04 | 14.22 $\pm$ 1.35     | 5:22            | 236            |
> | CP-AER        | 2.20e-03 $\pm$ 4.38e-05  | 95.61 $\pm$ 0.26    | 4.69e-02 $\pm$ 8.18e-04 | 95.60 $\pm$ 0.27     | 5:00            | 42             |
> | CP-PRE (Ours) | 2.20e-03 $\pm$ 4.96e-03  | 95.54 $\pm$ 0.18    | 4.71e-02 $\pm$ 1.06e-03 | 95.67 $\pm$ 0.22     | 5:00            | 82             |

---

> > ### Author Response · Authors · 2024-11-22
> > **Table Index**
> >
> > **Deterministic:** Vanilla FNO [1]
> >
> > **MC Dropout:** FNO with Dropout [2]
> >
> > **Deep Ensemble:** Ensemble of FNOs [3]
> >
> > **SWA-G:** Stochastic Weighted Averaging - Gaussian [4]
> >
> > **in-dist:** Model evaluated on initial states sampled from the same parameter range (as given in the appendix) of the initial condition as used in the training data.
> >
> > **out-dist:** Model evaluated on initial states sampled from a different parameter range of the initial conditions as used in the training data.
> >
> > **L2:** L2 norm of the model output with the ground truth in the normalised domain.
> >
> > **Coverage:** Percentage coverage of the model outputs within the estimated error bounds
> >
> > **Train Time:** Training time on a single A100 GPU.
> >
> > **Eval. Time:** Evaluation time on a single A100 GPU.
> >
> >
> > [1] Z. Li, N. B. Kovachki, K. Azizzadenesheli, B. Liu, K. Bhattacharya, A. Stuart, and A. Anandkumar, ``Fourier Neural Operator for Parametric Partial Differential Equations,'' in International Conference on Learning Representations (ICLR), 2021
> >
> > [2] Y. Gal and Z. Ghahramani, ``Dropout as a Bayesian Approximation: Representing Model Uncertainty in Deep Learning,'' in Proceedings of The 33rd International Conference on Machine Learning (ICML), vol. 48, pp. 1050--1059, PMLR, New York, USA, Jun. 2016
> >
> > [3] B. Lakshminarayanan, A. Pritzel, and C. Blundell, ``Simple and Scalable Predictive Uncertainty Estimation using Deep Ensembles,'' in Advances in Neural Information Processing Systems (NeurIPS), vol. 30, Curran Associates, Inc., 2017
> >
> > [4] M. McCabe, P. Harrington, S. Subramanian, and J. Brown, ``Towards Stability of Autoregressive Neural Operators,'' Transactions on Machine Learning Research, 2023

---

> > > ### Comment · Reviewer_YEc3 · 2024-11-25
> > >
> > > Thank you for your rebuttal. However, my concerns remain. I would like to further elaborate on them to ensure clarity:
> > >
> > > 1. The introduction of the PRE metric within the CP framework appears somewhat straightforward and lacks substantial novelty in addressing the underlying challenges.
> > >
> > > 2. I fully understand how you compute the PRE. However, on a coarsely discretized grid, the approximation of derivatives using finite differences introduces significant noise, which fails to accurately reflect the model’s approximation of the true PDE solution. This issue is particularly pronounced in the temporal dimension, where uncontrolled time intervals exacerbate the errors. Using such a noisy metric for uncertainty quantification raises questions about its practical utility.
> > >
> > > 3. In real-world scenarios, we often lack knowledge of the exact form of the underlying PDEs, rendering this method inapplicable in such cases.

---

> > > > ### Author Response · Authors · 2024-11-25
> > > > **The introduction of the PRE metric within the CP framework appears somewhat straightforward and lacks substantial novelty in addressing the underlying challenges.**
> > > >
> > > > We thank the reviewer for the quick responses and appreciate the efforts that they have put into helping us improve the paper. We have tried to accommodate the comments below.
> > > >
> > > > Although introducing a differential equation into the non-conformity score is conceptually and technically simple, producing uncertainty sets for the predictor in this setup non-trivial. This additionally gives:
> > > >
> > > > 1. A non-conformity score tailored to the problem at-hand based on physics, rather than usual fairly arbitrarily chosen scores.
> > > > 2. No longer needing to generate a calibration dataset from the numerical solver (a resource heavy part of current CP methods)
> > > > 3. Method to identify regions within the spatio-temporal domain as well as the parameter exploration space, where the physical inconsistencies are above a certain threshold.
> > > >
> > > > These are very desirable features of a PDE surrogate model to have, and we believe this constitutes an significant advancement over current conformal prediction methods.

---

> > > > ### Author Response · Authors · 2024-11-25
> > > > **Role of discretisation error in PRE-CP**
> > > >
> > > > We want to thank the reviewer for allowing us to explain the role discretisation plays within our method. Discretisation error is a common feature of numerical modelling and extends to neural operators as well
> > > > [1], [2] . Being autoregressive, we often do not have control over the temporal discretisation of the model as it is a feature of the training data and regime. This is even more pronounced when we utilise pre-trained foundation physics models [3], [4] with fixed temporal roll-outs. This leads to a larger discretisation error when estimating the PRE, however, we do not think that limits the utility of our method but presents more of a point of introspection, similar to the comment raised by reviewer mcYW.
> > > >
> > > > The discretisation error that emerges is a function of the PRE operator that we deploy and its effects are the same whether applied across the model output or the numerical simulation data (ground truth). It does not contribute to any noise but enforces features characteristic to the discretisation within the UQ method, mostly leading to scaled-up measures of the residual error. Using a coarser discretisation, our method still gives guaranteed coverage, but it will give wider error bars. As we increase the resolution of the output (spatially or temporally), our PRE estimates get smaller and we can get tighter error bounds. In Figure 10 of the revised submission (in Appendix D), we compare the PRE obtained across a coarsely discretised model prediction and a finely discretised model prediction and demonstrate the width of the error bars. Figure 10 (c) shows that irrespective of the discretisation, we obtain guaranteed coverage. The expected width of a particular PRE discretisation can be evaluated by applying it to a single numerical simulation with the same discretisation, similar to testing for model quality using PRE-CP as indicated in Appendix K. We could also explore the idea of bounding the discretisation error, but consider it out of scope for this current work [5].
> > > >
> > > > [1] F. Bartolucci, E. de Bezenac, B. Raonic, R. Molinaro, S. Mishra, and R. Alaifari, ``Representation Equivalent Neural Operators: a Framework for Alias-free Operator Learning,'' in Proceedings of the 37th Conference on Neural Information Processing Systems (NeurIPS), 2023
> > > >
> > > > [2] S. Lanthaler, A. M. Stuart, and M. Trautner, ``Discretization Error of Fourier Neural Operators,'' arXiv:2405.02221 [math.NA], 2024
> > > >
> > > > [3] M. McCabe, B. Régaldo-Saint Blancard, L. H. Parker, R. Ohana, M. Cranmer, A. Bietti, M. Eickenberg, S. Golkar, G. Krawezik, F. Lanusse, M. Pettee, T. Tesileanu, K. Cho, and S. Ho, ``Multiple Physics Pretraining for Physical Surrogate Models,'' arXiv:2310.02994 [cs.LG], 2023.
> > > >
> > > > [4] B. Alkin, A. Fürst, S. L. Schmid, L. Gruber, M. Holzleitner, and J. Brandstetter, ``Universal Physics Transformers: A Framework For Efficiently Scaling Neural Operators,'' in Proc. 38th Conference on Neural Information Processing Systems (NeurIPS), 2024.
> > > >
> > > > [5] Abarbanel, S., Ditkowski, A. & Gustafsson, B. On Error Bounds of Finite Difference Approximations to Partial Differential Equations—Temporal Behavior and Rate of Convergence. Journal of Scientific Computing 15, 79–116 (2000). https://doi.org/10.1023/A:1007688522777

---

> > > > ### Author Response · Authors · 2024-11-25
> > > > **In real-world scenarios, we often lack knowledge of the exact form of the underlying PDEs, rendering this method inapplicable in such cases.**
> > > >
> > > > The reviewer is correct in stating that this method is limited to scenarios where we have knowledge of the underlying PDE. However, we would like to humbly argue that modelling known PDEs is a crucial component of scientific computing, finding applications in weather modelling [1], modelling building fires [2] and computational fluid dynamics, cornerstone toolkit used in designing cars, aeroplanes and even nuclear reactors [3]. Considering the computational costs of performing these numerical simulations, ML-driven surrogate models have found utility in PDE modelling and have gained popularity in engineering design, predictive control and digital twinning [4], [5]. Quantifying the uncertainty of surrogate models with PRE in this context is crucial, as the degree of PDE violation gives confidence to the downstream user about the usability of the surrogate model. We would also like to take this opportunity to reiterate that our framework can easily be extended to ODEs or any other system where the model error could be expressed as a residual. Though the method fails when we are unable to formulate a residual for the problem, we would like to argue that there are plenty of safety-critical scenarios where this method could be applied and, hence is worth consideration for publication.
> > > >
> > > > [1]   Walters, D., Baran, A. J., Boutle, I., Brooks, M., Earnshaw, P., Edwards, J., Furtado, K., Hill, P., Lock, A., Manners, J., Morcrette, C., Mulcahy, J., Sanchez, C., Smith, C., Stratton, R., Tennant, W., Tomassini, L., Van Weverberg, K., Vosper, S., Willett, M., Browse, J., Bushell, A., Carslaw, K., Dalvi, M., Essery, R., Gedney, N., Hardiman, S., Johnson, B., Johnson, C., Jones, A., Jones, C., Mann, G., Milton, S., Rumbold, H., Sellar, A., Ujiie, M., Whitall, M., Williams, K., and Zerroukat, M.: The Met Office Unified Model Global Atmosphere 7.0/7.1 and JULES Global Land 7.0 configurations, Geosci. Model Dev., 12, 1909–1963, https://doi.org/10.5194/gmd-12-1909-2019, 2019.
> > > >
> > > > [2]  McGrattan, K. , McDermott, R. , Weinschenk, C. and Forney, G. (2013), Fire Dynamics Simulator, Technical Reference Guide, Sixth Edition, Special Publication (NIST SP), National Institute of Standards and Technology, Gaithersburg, MD, [online], https://doi.org/10.6028/NIST.sp.1018 (Accessed November 25, 2024)
> > > >
> > > > [3]  H. G. Weller, G. Tabor, H. Jasak, C. Fureby, A tensorial approach to computational continuum mechanics using object-oriented techniques, COMPUTERS IN PHYSICS, VOL. 12, NO. 6, NOV/DEC 1998.
> > > >
> > > > [4] Karniadakis, G.E., Kevrekidis, I.G., Lu, L. et al. Physics-informed machine learning. Nat Rev Phys 3, 422–440 (2021). https://doi.org/10.1038/s42254-021-00314-5
> > > >
> > > > [5] Brunton, S.L., Kutz, J.N. Promising directions of machine learning for partial differential equations. Nat Comput Sci 4, 483–494 (2024). https://doi.org/10.1038/s43588-024-00643-2

---

> ### Author Response · Authors · 2024-11-27
>
> We thank the reviewer for the helpful comments and the opportunity for further discussion. As we are approaching the last day allowed for PDF revision, could you kindly let us know if there are any other concerns that we should address?

---

### Official Review · Reviewer_U71S · 2024-11-01

**Soundness:** 1
**Presentation:** 2
**Contribution:** 1
**Rating:** 3
**Confidence:** 3

**Summary:**

The paper proposed to use a conformal prediction (CP) framework to estimate uncertainty in neural PDEs. The main proposal of the paper is the utilisation of physics residual errors as nonconformity scores within a CP framework. The main goal of the paper is to provide a fast uncertainty quantification framework for neural PDEs. The proposed method is illustrated on several toys examples.

**Strengths:**

If demonstrated more extensively through additional experiments and theoretical results, the paper potentially provides a computationally cheap uncertainty quantification framework for neural PDEs.

**Weaknesses:**

- At the current moment, the paper provides only a minor extension of the CP framework to the context of neural PDEs.
- There are no theoretical results in the paper supporting claims about bounds and statistical guarantees.
- The experiments are limited to several toy problems with only illustrative examples (and no simulation study with diverse/relevant evaluation metrics, such as l2 measure for the solution, coverage for the quality of CI, etc.).
- There is no comparison to any other uncertainty quantification methods.

**Questions:**

1)“The majority of literature in UQ for neural PDE solvers has been looking at Bayesian
methods, such as dropout, Bayesian neural networks and Monte Carlo methods (Geneva & Zabaras, 2020; Zou et al., 2024; Psaros et al., 2023), which lack guarantees or are computationally expensive.”
This statement is rather high-level. The authors should specify what guarantees these methods do not provide which conformal prediction does provide.
2) The paper does not distinguish between epistemic and aleatoric uncertainties and should be more explicit and precise about what type of uncertainty quantification it provides.
3) On line 155 there is a claim: “We introduce a novel data-free nonconformity score for Conformal Prediction (CP) to obtain statistically valid and guaranteed error bounds for neural PDE solvers.”
Theoretical support should be provided to this claim.
4) Both Figure 4 and Figure 5 illustrate very large confidence intervals for joint CP which would not correspond to 95% coverage. For marginal CP, it is hard to visually tell what coverage is. The same holds for the later figures. The paper should provide additional metrics to evaluate the results quantitatively.
5) Figure 4 and 5 provide a single illustrative example and not a proper Monte Carlo study to illustrate the calibration and coverage of the method. Considering that the paper suggests the method is computationally cheap, there is no reason to not provide more extensive results supporting the paper's claims.
6) There is no comparison to any other uncertainty quantification methods.

---

> ### Author Response · Authors · 2024-11-22
>
> Q: “The majority of literature in UQ for neural PDE solvers has been looking at Bayesian methods, such as dropout, Bayesian neural networks and Monte Carlo methods (Geneva & Zabaras, 2020; Zou et al., 2024; Psaros et al., 2023), which lack guarantees or are computationally expensive.”
>
> A: Thank you for the opportunity to clarify our statement regarding uncertainty quantification methods in neural PDE solvers. The primary methods - MC Dropout, Bayesian Neural Networks, and Deep Ensembles - each present significant practical challenges despite their theoretical appeal. MC Dropout requires numerous forward passes for reliable uncertainty estimates, which introduces substantial computational overhead during inference. Bayesian Neural Networks, while theoretically well-founded, demand extensive computational resources for MCMC or variational inference sampling and significantly increase memory requirements due to maintaining weight distributions. Deep Ensembles, though conceptually straightforward, necessitate training and storing multiple complete models, with costs scaling linearly with ensemble size. All these methods, being approximate Bayesian methods share common limitations: they lack theoretical guarantees on uncertainty coverage (especially for out-of-distribution samples), require extensive sampling and modifications to the model or training regime.  These challenges become particularly acute in real-world applications where both computational efficiency and reliable uncertainty estimates are crucial. We appreciate the reviewer's attention to this point and would be happy to provide additional technical details if needed.

---

> ### Author Response · Authors · 2024-11-22
> **Aleoteric vs Epistemic Uncertainty**
>
> Q: The paper does not distinguish between epistemic and aleatoric uncertainties and should be more explicit and precise about what type of uncertainty quantification it provides.
>
> A: We thank the reviewer for this question, it is the kind of uncertainty that conformal prediction characterises. From one perspective, conformal prediction can be viewed as aleatoric uncertainty, since we construct confidence intervals with respect to a specific probability distribution (the distribution the calibration data is drawn from). However, one could similarly argue the uncertainty is epistemic, since we are model the neural network's error, and use confidence intervals to-do so (used for scalar but unknown quantities, and therefore can be seen as Epistemic). The author's feeling is the second.
>
> The characterisation may not be useful with regards to this paper. As far as we understand, the characterisation is most useful when both types of uncertainty is present in the same problem (so that they don't become confounded, and that appropriate methods are used [1]). Perhaps this could be elaborated in the main text, however since this paper is not on fundamental statistics or fundamental Uncertainty Quantification, we feel the discussion is perhaps slightly out of scope.
>
> [1] S. Ferson, and L. Ginzburg,  "Different methods are needed to propagate ignorance and variability", Reliability Engineering \& System Safety, Elsevier, 1996, https://doi.org/10.1016/S0951-8320(96)00071-3

---

> ### Author Response · Authors · 2024-11-22
> **Theorem**
>
> Q: On line 155 there is a claim: “We introduce a novel data-free nonconformity score for Conformal Prediction (CP) to obtain statistically valid and guaranteed error bounds for neural PDE solvers.” Theoretical support should be provided to this claim.
>
> A: We thank the reviewer for this comment and at this opportunity we would like to highlight that our work extends from the principles of CP, where we extend the coverage guarantees to the residual space. Our work reformulates the prediction sets to be input-independent, allowing us to perform data-free CP. We had explored this by mathematical construction in appendix A, but valuing the feedback, we have formulated this into a Theorem. We have added this to the appendix of our paper.
>
>
> Preliminaries:
>
> Let $D: \mathbb{R}^m \rightarrow \mathbb{R}^m$ be a physics residual operator mapping a function to its PDE residual value, where:  $\\{X_i\\}_{i=1}^n$ is the calibration set, $\hat{f}$ is the model, $\hat{q}^\alpha$ is estimated as the $\lceil(n+1)(1-\alpha)\rceil/n$ -quantile of
>
> $\\{|D(\hat{f}(X_i))|\\}_{i=1}^n$
>
>
> Theorem:
>
> If the residuals $\\{D(\hat{f}(X_i))\\}_{i=1}^{n+1}$
>
> are exchangeable random variables, then for any significance level $\alpha \in (0,1)$ and any new input $X{n+1}$ we have the following coverage guarantee:
> $$\mathbb{P}(|\mathcal{D}(\hat{f}(X_{n+1}))| \in C_\alpha) \geq 1 - \alpha , \\;   C_\alpha = [-\hat{q}\alpha, \hat{q}\alpha]$$
>
> Proof:
>
> Let $R_i = |D(\hat{f}(X_i))|$ for $i=1,\ldots,n+1$. We have, by assumption, $(R_1,\ldots,R_n,R_{n+1})$ is an exchangeable sequence. Define the rank $\pi$ of $R_{n+1}$ w.r.t. all other residuals:
> $$\pi(R_{n+1}) = |{i=1,\ldots,n+1: R_i \leq R_{n+1}}|$$
> By exchangeability, the rank $\pi(R_{n+1})$ is uniformly distributed over ${1,\ldots,n+1}$. Therefore,
> $$P(\pi(R_{n+1}) \leq \lceil(n+1)(1-\alpha)\rceil) = \frac{\lceil(n+1)(1-\alpha)\rceil}{n} \geq 1-\alpha.$$
> By construction of $\hat{q}^\alpha$ we have that,
> $${\pi(R_{n+1}) \leq \lceil(n+1)(1-\alpha)\rceil} \subseteq {R_{n+1} \leq \hat{q}^\alpha}.$$
> Putting this together,
> $$P(|D(\hat{f}(X_{n+1}))| \leq \hat{q}^\alpha) = P(R_{n+1} \leq \hat{q}^\alpha) \geq 1-\alpha,$$
> which completes the proof. ■

---

> ### Author Response · Authors · 2024-11-22
>
> Q: Both Figure 4 and Figure 5 illustrate very large confidence intervals for joint CP which would not correspond to 95% coverage. For marginal CP, it is hard to visually tell what coverage is. The same holds for the later figures. The paper should provide additional metrics to evaluate the results quantitatively.
>
> A: We thank the reviewer for this comment. Visualising coverage for marginal CP is difficult as it is done individually for each cell in the spatio-temporal output tensor. In figures 4 and 5 it is shown for a single example at a fixed point in time. Figure 2 checks whether coverage is valid for each error cell and averages across the domain. To provide further clarity, while comparing against other UQ measures, we showcase the coverage obtained for $\alpha=0.05$, averaged across different samples of the inputs along with the l2 fit of the model, both within and out of distribution cases while comparing against other UQ methods in Tables 2-4 within the appendix section C.

---

> ### Author Response · Authors · 2024-11-22
>
> Q: Figure 4 and 5 provide a single illustrative example and not a proper Monte Carlo study to illustrate the calibration and coverage of the method. Considering that the paper suggests the method is computationally cheap, there is no reason to not provide more extensive results supporting the paper's claims.
>
> A: Thank you for this comment, but we believe there is a slight misunderstanding. Although the reviewer is correct that Figures 4. and 5. show single examples, a detailed Monte Carlo validation of our method \textit{has} been performed. It is what Figure 3 of the main text shows. This figure shows how the empirical coverage, the true coverage computed using Monte Carlo simulation, changes as we change the $\alpha$-level of our constructed confidence intervals. This is an empirical / Monte Carlo validation of equation (5) of the main paper (marginal coverage of conformal prediction). While comparing against other UQ methods in tables 2-4, we conduct further coverage checks by performing MC sampling across a larger dataset.

---

> ### Author Response · Authors · 2024-11-22
> **Comparison to other UQ methods**
>
> We did not include comparisons to other UQ methods as our method involved quantifying the physics residual error and obtaining its bounds as opposed to directly quantifying the model's predictive error. However, considering the concerns raised by the reviewer regarding providing baseline comparisons and the effectiveness such a comparison provides to our claims, we have conducted experiments to demonstrate how our method stands out as being data-free, not requiring any modifications or sampling, and helps obtain guaranteed coverage bounds in a physics-informed manner. We have added them to the revised paper under Appendix C.

---

> > ### Author Response · Authors · 2024-11-22
> > **Qualitative Comparison**
> >
> > | Method        | Data-Free | Modification-Free | Sampling-Free | Guaranteed Coverage | Physics-Informed |
> > |---------------|-----------|-------------------|---------------|---------------------|------------------|
> > | MC Dropout    | ✓         | ✗                 | ✗             | ✗                   | ✗                |
> > | Deep Ensemble | ✓         | ✗                 | ✗             | ✗                   | ✗                |
> > | BNN           | ✓         | ✗                 | ✗             | ✗                   | ✗                |
> > | SWA-G         | ✓         | ✗                 | ✗             | ✗                   | ✗                |
> > | CP-AER        | ✗         | ✓                 | ✓             | ✓                   | ✗                |
> > | CP-PRE (Ours) | ✓         | ✓                 | ✓             | ✓                   | ✓                |
> >
> >  Our method is data-free, does not require any modifications or sampling, and helps obtain guaranteed coverage bounds in a physics-informed manner.

---

> ### Author Response · Authors · 2024-11-22
> **Additional Experiments**
>
> Though we disagree with the reviewer's comments that our work only touches on toy problems, we are thrilled at the opportunity to demonstrate further experiments of our method to a more complex scenario.
>
> In [1], the authors model the evolution of plasma blobs within a Fusion reactor (known as a Tokamak) following reduced magnetohydrodynamics. Plasma, characterised by density $\rho$, electric potential $\phi$ and Temperature $T$ under the absence of magnetic pressure confining it in place, moves radially outward to the wall of the reactor driven by its kinetic pressure.  We demonstrate the ability to scale our method by applying it to obtain valid error bars across the multi-variable FNO trained for plasma modelling.
>
> As shown in figure  23 of the revised paper, our method can capture the model error across a range of predictions and can devise error bars that provide guaranteed coverage without needing any additional data. In figure 23(a), we demonstrate the absolute error in the model prediction of the temperature evolution, and correlate that with the PRE over the temperature equations in figure 23(b). The details of the new experiment has been added to Appendix L of the revised paper.
>
> [1] V. Gopakumar, S. Pamela, L. Zanisi, Z. Li, A. Gray, D. Brennand, N. Bhatia, G. Stathopoulos, M. Kusner, M. P. Deisenroth, A. Anandkumar, the JOREK Team, and MAST Team, "Plasma surrogate modelling using Fourier neural operators," Nuclear Fusion, vol. 64, no. 5, p. 056025, Apr. 2024, doi: 10.1088/1741-4326/ad313a

---

> ### Author Response · Authors · 2024-11-22
> **Quantitative comparison to other UQ methods**
>
> ## `Wave Equation - Coverage measured for 2σ(∼ 95%)`
>
> #
>
> | UQ            | L2 (in-dist.)           | Coverage (in-dist.) | L2 (out-dist.)          | Coverage (out-dist.) | Train time (hr) | Eval. time (s) |
> |:-------------:|:-----------------------:|:-------------------:|:-----------------------:|:--------------------:|:---------------:|:--------------:|
> | Deterministic | 1.77e-05 $\pm$ 3.69e-07 | -                   | 2.46e-03 $\pm$ 2.00e-05 | -                    | 0:38            | 22             |
> | MC Dropout    | 1.44e-04 $\pm$ 3.26e-06 | 97.31 $\pm$ 0.03    | 2.12e-03 $\pm$ 2.60e-05 | 89.83 $\pm$ 0.07     | 0:52            | 120            |
> | Deep Ensemble | 8.76e-06 $\pm$ 2.43e-07 | 98.02 $\pm$ 0.04    | 2.42e-03 $\pm$ 1.58e-05 | 83.44 $\pm$ 0.12     | 3:10            | 112            |
> | BNN           | 1.92e-04 $\pm$ 1.92e-06 | 97.10 $\pm$ 0.09    | 2.67e-03 $\pm$ 1.26e-05 | 91.76 $\pm$ 0.10     | 0:53            | 118            |
> | SWA-G         | 1.41e-05 $\pm$ 1.74e-06 | 94.55 $\pm$ 3.25    | 2.55e-03 $\pm$ 2.82e-05 | 81.90 $\pm$ 3.31     | 0:47            | 113            |
> | CP-AER        | 1.76e-05 $\pm$ 4.40e-07 | 95.70 $\pm$ 0.21    | 2.46e-03 $\pm$ 1.41e-05 | 95.59 $\pm$ 0.14     | 0:38            | 23             |
> | CP-PRE (Ours) | 1.78e-05 $\pm$ 4.61e-07 | 95.52 $\pm$ 0.21    | 2.46e-03 $\pm$ 1.25e-05 | 95.39 $\pm$ 0.12     | 0:38            | 23             |
>
> #
> ## `Navier-Stokes Equations - Coverage measured for 2σ(∼ 95%)`
>
> #
>
>
> | UQ            | L2 (in-dist.)           | Coverage (in-dist.) | L2 (out-dist.)           | Coverage (out-dist.) | Train time (hr) | Eval. time (s) |
> |:-------------:|:-----------------------:|:-------------------:|:------------------------:|:--------------------:|:---------------:|:--------------:|
> | Deterministic | 1.05e-04 $\pm$ 6.91e-06 | -                   | 3.67e-03 $\pm$ 5.30e-05  | -                    | 3:22            | 25             |
> | MC Dropout    | 5.96e-04 $\pm$ 2.30e-05 | 82.21 $\pm$ 0.22    | 4.30e-03 $\pm$ 8.05e-05  | 44.05 $\pm$ 0.26     | 3:34            | 153            |
> | Deep Ensemble | 1.22e-04 $\pm$ 3.95e-06 | 91.31 $\pm$ 0.08    | 3.67e-03 $\pm$ 3.52e-05  | 30.74 $\pm$ 0.19     | 16:22           | 147            |
> | BNN           | 6.90e-03 $\pm$ 1.31e-04 | 89.91 $\pm$ 0.20    | 6.95e-03 $\pm$  1.31e-04 | 85.19 $\pm$ 0.23     | 3:39            | 152            |
> | SWA-G         | 1.96e-04 $\pm$ 1.15e-05 | 84.22 $\pm$ 2.37    | 3.63e-03 $\pm$ 1.37e-04  | 31.00 $\pm$ 2.85     | 3:28            | 146            |
> | CP-AER        | 1.05e-04 $\pm$ 6.58e-06 | 95.56 $\pm$ 0.40    | 3.66e-03 $\pm$ 2.81e-05  | 95.54 $\pm$ 0.15     | 3:22            | 26             |
> | CP-PRE (Ours) | 1.07e-04 $\pm$ 5.18e-06 | 95.44 $\pm$ 0.22    | 3.70e-03 $\pm$ 4.23e-05  | 95.57 $\pm$ 0.14     | 3:22            | 34             |          |         |           |          |      |       |      |
>
>
> #
> ## `MHD Equations - Coverage measured for 2σ(∼ 95%)`
>
> #
> | UQ            | L2 (in-dist.)            | Coverage (in-dist.) | L2 (out-dist.)          | Coverage (out-dist.) | Train time (hr) | Eval. time (s) |
> |:-------------:|:------------------------:|:-------------------:|:-----------------------:|:--------------------:|:---------------:|:--------------:|
> | Deterministic | 2.20e-03 $\pm$ 5.20e-03  | -                   | 4.71e-02 $\pm$ 1.06e-03 | -                    | 5:00            | 40             |
> | MC Dropout    | 3.29e-02 $\pm$ 5.86e-04  | 41.13 $\pm$ 0.19    | 2.09e-01 $\pm$ 1.38e-03 | 16.91 $\pm$ 0.06     | 5:30            | 240            |
> | Deep Ensemble | 3.59e-03 $\pm$ 3.51e-04  | 78.15 $\pm$ 0.16    | 3.41e-01 $\pm$ 3.15e-02 | 39.63 $\pm$ 0.31     | 26:25           | 235            |
> | BNN           | 4.20e-03 $\pm$  4.08e-05 | 90.24 $\pm$ 0.10    | 4.63e-02 $\pm$ 8.98e-04 | 62.37 $\pm$ 0.46     | 5:40            | 240            |
> | SWA-G         | 2.61e-03 $\pm$ 9.68e-05  | 48.50 $\pm$ 3.81    | 4.53e-02 $\pm$ 6.64e-04 | 14.22 $\pm$ 1.35     | 5:22            | 236            |
> | CP-AER        | 2.20e-03 $\pm$ 4.38e-05  | 95.61 $\pm$ 0.26    | 4.69e-02 $\pm$ 8.18e-04 | 95.60 $\pm$ 0.27     | 5:00            | 42             |
> | CP-PRE (Ours) | 2.20e-03 $\pm$ 4.96e-03  | 95.54 $\pm$ 0.18    | 4.71e-02 $\pm$ 1.06e-03 | 95.67 $\pm$ 0.22     | 5:00            | 82             |          |         |           |          |      |       |      |

---

> > ### Author Response · Authors · 2024-11-22
> > **Table Index**
> >
> > **Deterministic:** Vanilla FNO [1]
> >
> > **MC Dropout:** FNO with Dropout [2]
> >
> > **Deep Ensemble:** Ensemble of FNOs [3]
> >
> > **SWA-G:** Stochastic Weighted Averaging - Gaussian [4]
> >
> > **in-dist:** Model evaluated on initial states sampled from the same parameter range (as given in the appendix) of the initial condition as used in the training data.
> >
> > **out-dist:** Model evaluated on initial states sampled from a different parameter range of the initial conditions as used in the training data.
> >
> > **L2:** L2 norm of the model output with the ground truth in the normalised domain.
> >
> > **Coverage:** Percentage coverage of the model outputs within the estimated error bounds
> >
> > **Train Time:** Training time on a single A100 GPU.
> >
> > **Eval. Time:** Evaluation time on a single A100 GPU.
> >
> >
> > [1] Z. Li, N. B. Kovachki, K. Azizzadenesheli, B. Liu, K. Bhattacharya, A. Stuart, and A. Anandkumar, ``Fourier Neural Operator for Parametric Partial Differential Equations,'' in International Conference on Learning Representations (ICLR), 2021
> >
> > [2] Y. Gal and Z. Ghahramani, ``Dropout as a Bayesian Approximation: Representing Model Uncertainty in Deep Learning,'' in Proceedings of The 33rd International Conference on Machine Learning (ICML), vol. 48, pp. 1050--1059, PMLR, New York, USA, Jun. 2016
> >
> > [3] B. Lakshminarayanan, A. Pritzel, and C. Blundell, ``Simple and Scalable Predictive Uncertainty Estimation using Deep Ensembles,'' in Advances in Neural Information Processing Systems (NeurIPS), vol. 30, Curran Associates, Inc., 2017
> >
> > [4] M. McCabe, P. Harrington, S. Subramanian, and J. Brown, ``Towards Stability of Autoregressive Neural Operators,'' Transactions on Machine Learning Research, 2023

---

> ### Author Response · Authors · 2024-11-27
>
> We thank the reviewer for the helpful comments and the opportunity for further discussion. As we are approaching the last day allowed for PDF revision, could you kindly let us know if there are any other concerns that we should address?

---

### Official Review · Reviewer_mcYW · 2024-11-03

**Soundness:** 3
**Presentation:** 4
**Contribution:** 4
**Rating:** 6
**Confidence:** 4

**Summary:**

This paper presents a data-free conformal prediction approach for uncertainty quantification in neural PDE solvers, which does not require additional true PDE solutions to evaluate nonconformity scores. This data-free capability is achieved by assessing physics residual errors (PRE), or violations of conservation laws, for each prediction. PRE-CP provides both marginal (dimension-wise) and joint (domain-wide) coverage, enabling statistically sound conformal bounds for predictions. Additionally, the authors employ convolutional layers as finite difference stencils for faster and more accurate gradient estimation, as well as a rejection sampling method that uses CP-based criteria to reject predictions with a predetermined probability.

**Strengths:**

•	Very interesting and important topic for SciML community and very well written

•	Strong motivation

•	Complete experiments and discussion with strong empirical results

•	Methods are clearly written and are easy to follow

**Weaknesses:**

I have some question regarding how will PRE-CP behave when the neural model fails to predict PDE solution, or generating consistently high physical residual errors.

**Questions:**

Krishnapriyan et al. [1] show that it is highly likely for PINNs to fail in predicting PDE solutions when the PDE coefficients are high, which can lead to consistently high residual errors across almost the entire PDE domain. I am curious whether PRE-CP would generate very wide conformal bounds that still provide relatively high coverage of these residual errors, potentially underestimating the poor performance of the neural solver.

[1] Krishnapriyan, Aditi, et al. "Characterizing possible failure modes in physics-informed neural networks." Advances in neural information processing systems 34 (2021): 26548-26560.

---

> ### Author Response · Authors · 2024-11-22
> **PRE-CP when the neural model fails to predict the PDE solution.**
>
> We thank the reviewer for this question, which touches upon an important aspect of neural PDEs: their fit not just to the data but to the underlying physics. The reviewer is spot-on in assuming that PRE-CP will still provide guaranteed coverage with considerably wider error bounds when the neural-PDE (whether PINN or a Neural Operator) fails to comply with the physics. However, we believe that this is an advantage of our method. In PRE-CP formulation, the bounds are estimated across the PDE residual, where the ground truth for a well-fit solution should always be near zero. If we get wide error bars further away from the 0 for potentially high coverage estimates, it is a strong indication that statistically the model violates the physics of interest.
>
> Consider the example with the Advection equation. We have two models, a well-fit (good model) and a poorly fit one (bad model). As shown in figure 22 in the revised paper, though we obtain guaranteed coverage in the case of both the bad and good models, the width of the error bars indicates the quality of the model. Taken for 90 \% coverage, the width of the coverage bounds obtained over the bad model is substantially larger than that obtained by the good model.
>
> There still could be a concern as to what width can be considered to be within a good range within the residual space. This could be estimated by running the PRE convolution operator(s) across a single numerical simulation of the interested physics, thereby estimating the impact of the operator in estimating the residual. The PRE over the simulation data will allow us to judge what ranges for the coverage width differentiate between a "bad" and a "good" model.  Thus, PRE can be utilised as a metric to evaluate the physics deviation, and as a nonconformity score to get bounds on the physics deviation. We have added this capability to the revised paper under appendix K.

---

> > ### Comment · Reviewer_mcYW · 2024-11-26
> >
> > Thank you for your explanation. I now understand it. Thank you.

---

### Meta-Review · Area_Chair_LwB8 · 2024-12-16

**Metareview:**

The paper presents a data-free conformal prediction approach for uncertainty quantification in neural PDE solvers, which does not require additional true PDE solutions to evaluate nonconformity scores. The authors had tried to address the reviewers' comments and questions. Some reviewers still have some concerns and do not want to support the publication of the paper. The reviewer thinks that the method is limited in practice as we often lack knowledge of the exact form of the underlying PDEs. Moreover, the reviewer also mentioned that the method is somewhat straightforward and lacks substantial novelty in addressing the underlying challenges. Unfortunately, I cannot recommend the acceptance of this paper based on the reviewers' recommendations. I strongly encourage the authors to clarify all the points that reviewers mentioned to improve the paper for the future submissions.

**Additional Comments On Reviewer Discussion:**

NA

---

### Decision · Program_Chairs · 2025-01-22

Reject